EMBO
Molecular Medicine

# Kidney cancer PDOXs reveal patient-specific pro-malignant effects of antiangiogenics and its molecular traits

Lidia Moserle[1,†‡], Roser Pons[1,†], Mar Martínez-Lozano[1], Gabriela A Jiménez-Valerio[1], August Vidal[2], Cristina Suárez[3], Enrique Trilla[4], José Jiménez[3], Inés de Torres[5], Joan Carles[3], Jordi Senserrich[1], Susana Aguilar[1], Luis Palomero[6], Alberto Amadori[7,8] & Oriol Casanovas[1,*]

## Abstract

An open debate in antiangiogenic therapies is about their consequence on tumor invasiveness and metastasis, which is undoubtedly relevant for patients currently treated with antiangiogenics, such as renal cell carcinoma patients. To address, this we developed an extensive series of 27 patient biopsy-derived orthotopic xenograft models (Ren-PDOX) that represent inter-patient heterogeneity. In specific tumors, antiangiogenics produced increased invasiveness and metastatic dissemination, while in others aggressiveness remained unchanged. Mechanistically, species-discriminative RNA sequencing identified a tumor cell-specific differential expression profile associated with tumor progression and aggressivity in TCGA RCC patients. Gene filtering using an invasion-annotated patient series pinpointed two candidate genes, of which ALDH1A3 differentiated the pro-invasive subtype of Ren-PDOXs. Validation in an independent series of 15 antiangiogenic-treated patients confirmed that pre-treatment ALDH1A3 can significantly discriminate patients with pro-aggressive response upon treatment. Overall, results confirm that effects of antiangiogenic drugs on tumor invasion and metastasis are heterogeneous and may profoundly affect the natural progression of tumors and promote malignancy. Furthermore, we identify a specific molecular biomarker that could be used to select patients that better benefit from treatment.

**Keywords** antiangiogenics; biomarker; cancer resistance; metastasis induction; orthotopic models of kidney cancer
**Subject Categories** Cancer; Vascular Biology & Angiogenesis

## Introduction

Antiangiogenic therapies have been typically identified as promising therapies for treatment of several types of cancers sustained by angiogenesis to progress (Folkman, 2007). Since they have been introduced in the treatment of oncologic diseases, several patients have benefited from these drugs. Nevertheless, they have not been the expected panacea. Clinical studies indeed have shown that antiangiogenic treatments exert promising results in terms of prolonged progression-free survival (PFS) but eventually the efficacy decreases thus resulting in less clear effects on overall survival (OS) (Al-Husein *et al*, 2012).

Several studies in mouse models have tried to explain the discrepancy between promising short-term effects and the long-term limitations of antiangiogenic therapies and allowed finding out some possible mechanisms involved in failure (Bergers & Hanahan, 2008; Ellis & Hicklin, 2008; Moserle *et al*, 2014). Interestingly, several preclinical studies, including ours, demonstrated that antiangiogenic treatments were able to induce or exacerbate the invasive and metastatic behavior of different types of tumors (Ebos *et al*, 2009; Paez-Ribes *et al*, 2009; Sennino *et al*, 2012; Shojaei *et al*, 2012). On the contrary, other authors described controversial pro-invasive effects of anti-VEGF therapies that could depend on drug-specific side-effects on tumor microenvironment more than on the blockade of VEGF pathway (Chung *et al*, 2012; Singh *et al*, 2012). Hence, there is still an open question if and when antiangiogenic therapies can increase tumor invasiveness and unexpectedly modify the

1 Tumor Angiogenesis Group, ProCURE Program, Catalan Institute of Oncology, OncoBell Program, IDIBELL, Barcelona, Spain
2 Department of Pathology, University Hospital of Bellvitge, Bellvitge Biomedical Research Institute (IDIBELL), CIBERONC, Barcelona, Spain
3 Medical Oncology Department, Vall d'Hebron Hospital, Barcelona, Spain
4 Surgery Department, Vall d'Hebron Hospital, Barcelona, Spain
5 Pathology Department, Vall d'Hebron Hospital, Barcelona, Spain
6 ProCURE Program, Catalan Institute of Oncology. OncoBell Program, IDIBELL, Barcelona, Spain
7 Department of Surgery, Oncology and Gastroenterology, University of Padova, Padova, Italy
8 Veneto Institute of Oncology IOV – IRCCS, Padua, Italy
*Corresponding author. Tel: +34 932607463; E-mail: ocasanovas@iconcologia.net
†These authors contributed equally to this work
‡Current address: Veneto Institute of Oncology IOV – IRCCS, Padua, Italy

progression of neoplasia toward more aggressive behavior eventually explaining the marginal long-term efficacy.

Among human tumors, renal cell carcinoma (RCC) is commonly treated with antiangiogenics but gain in patient survival is controversial (Escudier et al, 2007; Rini et al, 2008; Motzer et al, 2009; Rousseau et al, 2016). Furthermore, when evaluated, the impact on metastatic progression is unclear (Helgason et al, 2008; Plimack et al, 2009; Massard et al, 2010; Miles et al, 2011; Vanhuyse et al, 2012). Doubt also exists for the safety of antiangiogenics administered as neoadjuvant therapy (Borregales et al, 2016). Renal tumors grow surrounded by a fibrous peritumoral capsule and capsular invasion in localized RCC has been suggested as a prognostic factor of recurrence and of cancer-related death (Cho et al, 2009; Snarskis et al, 2016). Intriguingly, a study in a small subset of RCC patients treated with the antiangiogenic tyrosine kinase inhibitor axitinib, as neoadjuvant therapy, has described a fibrous reaction in the tumor-normal kidney interface and an invasive proliferative pattern in most treated tumors (Kawakami et al, 2016). Indeed, anti-VEGF therapy produces ECM remodeling and pro-aggressive tumor behavior in other tumor types (Aguilera et al, 2014; Rahbari et al, 2016).

Patient-derived xenografts (PDXs) have recently become fundamental resources in translational studies. Maintaining human tumor biology and principal characteristics of the tumor of origin, they recapitulate the variability of human cancer and could therefore contribute to understand mechanisms behind positive therapy outcome in some patients but failure in others. Consequently, they may help to disclose predictive parameters for therapy response (Hidalgo et al, 2014; Gao et al, 2015). Specifically in orthotopic patient-derived model (PDOX), some parallelism exists between clinical results and preclinical observations (Hoffman, 2015), such as in glioblastoma where VEGF-targeted therapies failed instead inducing vessel cooption and tumor cell spread in patients and PDX models (Lamszus et al, 2005; Norden et al, 2008; Joo et al, 2013).

In renal cancer, published studies from our and other groups have recently demonstrated that PDOXs recapitulate the effects of antiangiogenic therapy on patients (Sivanand et al, 2012; Thong et al, 2014; Jimenez-Valerio et al, 2016) being good predictive models for therapy outcome. Can they also be a proper tool to investigate whether inhibition of VEGF pathway could promote RCC progression?

Here, in a "bedside to bench" approximation, we developed a significant collection of Ren-PDOX models and we found that antiangiogenics could specifically switch on tumor invasiveness and increase systemic dissemination of spontaneously low infiltrative tumors, but in another subset of Ren-PDOX models, aggressiveness remained unchanged after anti-VEGF/R therapies. In order to unravel the key differences between these two distinct responses, tumor cell-specific RNA sequencing profiling revealed that KRAS signaling could be involved in this divergent invasive behavior upon antiangiogenic treatment. Further TCGA aggressiveness filtering and validation in an invasion-annotated series of patients pinpointed a two-gene signature that could be used to discriminate pro-invasive from non-pro-invasive tumors. Together, the divergent effects of tumor malignization observed in individual Ren-PDOXs suggest that the controversy on pro-invasive effect of antiangiogenics may be explained by inter-patient heterogeneity, and our data define a two-gene signature as a putative prediction biomarker to discriminate pro-invasive vs non-pro-invasive RCC tumors in the clinics.

# Results

## 786-O orthoxenograft model of kidney invasion and lung metastasis

To define the characteristics of tumor invasion and metastasis, we first set up quantitative parameters of tumor invasiveness using an orthoxenograft RCC mouse model generated by orthotopic implantation of 3D tumor pieces derived from intrakidney injection of 786-O cells (Fig EV1A). The implanted tumor pieces grew bound to the kidney with defined invasive front at the tumor-kidney interface where the tumor edge was irregular with strands of cells going into renal parenchyma (Fig EV1B). 786-O tumors expressed high levels of human vimentin, a marker typically used for clear cell RCC (ccRCC) characterization (Truong & Shen, 2011) (Fig EV1B and C).

RCC patients typically develop lung metastases (Weiss et al, 1988). We therefore investigated the spontaneous lung metastatic potential of 786-O orthoxenografts macroscopically by direct observation of lungs at mice sacrifice and microscopically by vimentin staining of lung sections (Fig EV1D). In a few cases, we observed macrometastases while around 15% of mice developed micrometastases.

## Antiangiogenics exacerbate tumor local invasion and metastatic dissemination in 786-O orthoxenografts

In order to study the consequences of VEGF blockade on 786-O orthoxenograft tumors, we administered tumor-bearing mice with anti-VEGFR2 DC101 (DC) or anti-human VEGF bevacizumab (Beva) (Fig 1A). We found that DC and Beva treatments impaired tumor growth (Fig 1B and C) due to inhibition of angiogenesis as suggested or demonstrated by vessel number reduction and increased tumor necrosis (Fig 1D and E). Moreover, experiments until ethically accepted survival of each mouse demonstrated that DC and Beva treatments extended OS (Fig 1F). Nevertheless, by inspection of invasive front in vimentin-stained tumor sections, we observed strands of cells entering deeper into the renal parenchyma in treated mice compared to control tumors both at short and long-term treatment (Fig 1G and data not shown). Measuring local invasion, we demonstrated that tumor invasiveness significantly increased after both DC and Beva short-term treatments by >2-fold compared to basal invasion of untreated tumors (Fig 1G). The results suggest that although effective, anti-VEGF/R therapies may increase the invasive capacity of 786-O tumors as an early event.

To investigate the consequence of inhibition of VEGF signaling on metastasis, we macroscopically and microscopically analyzed lungs of treated and control mice in survival experiments at sacrifice. While we did not observe any macrometastases in the lung lobules, microscopic evaluation of lung tissue sections by vimentin immunostaining showed that the incidence of animals with micrometastasis was threefold and fourfold increased in DC and Beva-treated mice, respectively (Fig EV1D and H). Furthermore, the number of metastatic foci per mouse was significantly higher in both treated groups compared to controls (Fig 1H), whereas total area of lung tumor burden did not change (data not shown). These results were consistent with the increase of local invasion upon treatment and indicated that antiangiogenics could promote the systemic spreading of cells from primary tumor. Altogether, these

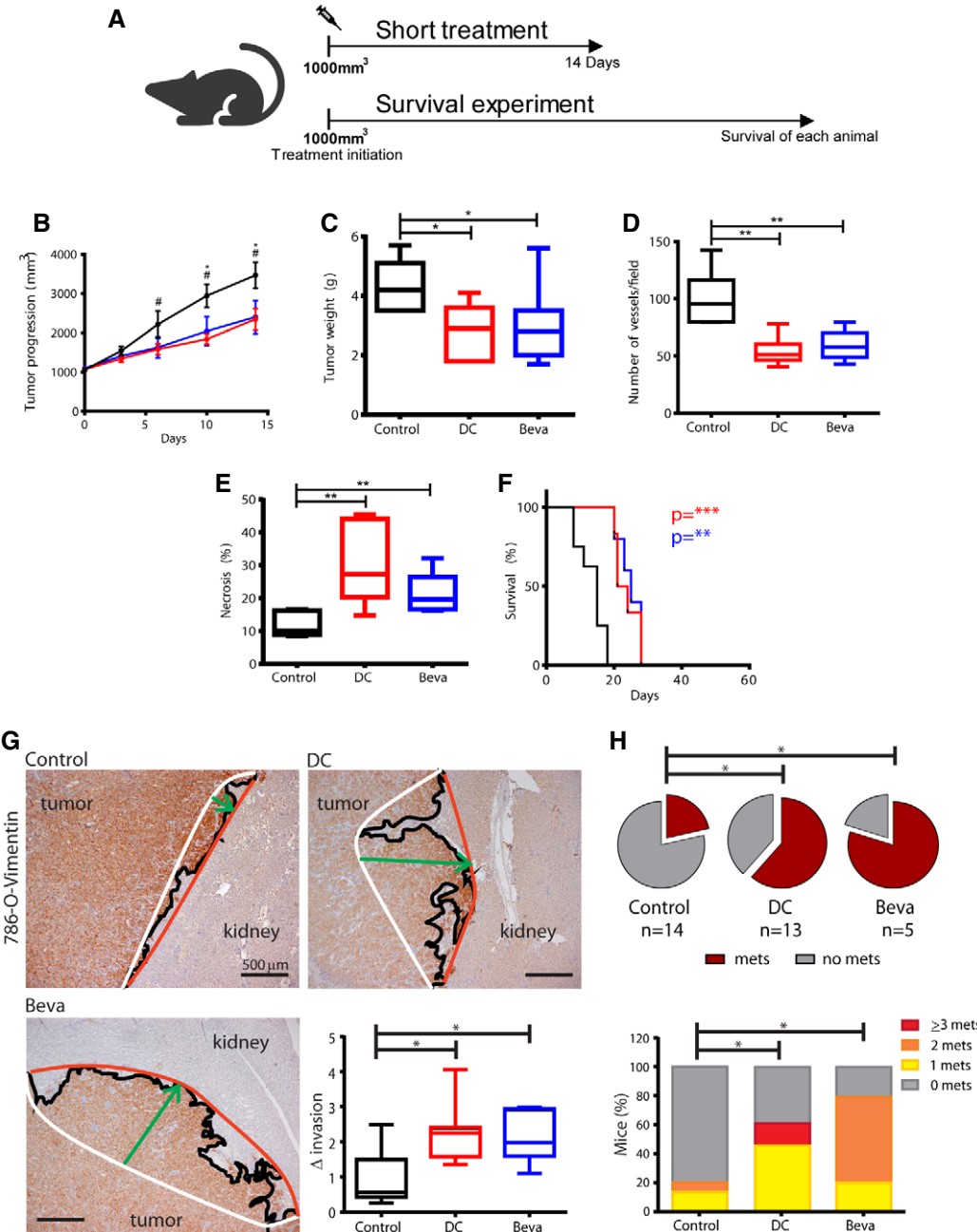

**Figure 1. Effects of anti-VEGF/R therapy on tumor progression, invasiveness, and metastatization in 786-O orthoxenograft.**

A   When tumor was palpable, mice were treated with two different antiangiogenics, DC and Beva. For short treatment experiments, the duration of treatment was 14 days. For survival experiments, therapies were administered until survival of each mouse.

B–F   Effects of DC and Beva treatment in 786-O orthoxenograft. Evaluation of (B) tumor progression (average ± SD), (C) tumor weight at sacrifice, (D) vessel number, (E) percentage of necrotic area by histology, and (F) survival by Kaplan–Meier plot in control and treated animals. Box plots indicate median, Q1/Q3 and max/min value whiskers in Control (black), DC101 (red), and Beva (blue) treatment groups. 5–8 tumors/group were analyzed by Mann–Whitney test, and Mantel-Cox test for survival where *$P < 0.05$, **$P < 0.01$ and ***$P < 0.001$ and DC *$P < 0.05$ and Beva #$P < 0.05$ vs. Control).

G   Representative invasive front profile of 786-O control and short-term DC and Beva-treated tumors in vimentin-stained sections (4X). Front (red line) and rear (white line) edges of tumor protrusion into kidney determine the depth of invasion (green arrow) perpendicularly connecting red and white lines. Box plot shows the quantification of invasion indicating median, Q1/Q3 and max/min value whiskers of fold-invasion (Δ invasion) in treated *vs* control tumors (seven samples/group; *$P < 0.05$ by Mann–Whitney test).

H   Evaluation of lung micrometastases in survival experiments. Incidence (pie chart, *$P < 0.05$ by Chi-square) and proportion of mice with 0, 1, 2, or ≥3 metastases (mets) per lungs (*$P < 0.05$ by Mann–Whitney test) are shown.

data suggest that in 786-O-derived orthoxenografts VEGF-targeted therapies could initially modify tumor invasive behavior and eventually result in enhanced systemic dissemination.

Although useful, this cell-based model has limited representativity of patient's genetic variability and heterogeneous response to therapy, as it derives from one single patient's cell line. Therefore, we aimed to provide with more relevant models to define patient's responses to therapy in the context of inter-patient genetic heterogeneity.

### Ren-PDOX models recapitulate architectural features and metastatic potential of their original patient's tumor

In order to evaluate the impact of antiangiogenic drugs in clinically relevant models, we generated Ren-PDOXs by direct implantation of human tumor specimens from patient into mouse kidney (Fig 2A). We collected and implanted a total of 56 ccRCC and we obtained 27 growing PDOX corresponding to 48.2% of engraftment (Appendix Tables S1 and S2). During the *in vivo* passage, Ren-PDOX maintained the same structural features of original human specimen, distinctive of each tumor (Fig 2B). As expected, by CD31 and CD34 IHC staining as species-specific markers of murine and human blood vessels respectively, we found that human stroma components were lost in PDOXs and substituted by murine stroma from the first orthotopic *in vivo* growth (Fig 2C). This was confirmed by species-specific TaqMan analysis of murine and human VEGF-A and VEGFR2 where mostly human VEGF-A ligand and only mouse VEGFR2 were detected (Fig EV2A and B).

Tumor harvested from metastatic patients (including positive regional lymph node and/or synchronous or metachronous metastasis) exhibited significantly higher engraftment rate compared to patients that did not present metastasis (pN0 and pM0), generating PDOX, respectively, in 88.9% and 28.1% of implanted tumors ($P < 0.0001$ by Chi-square test). Moreover, we found that 38.5% of growing Ren-PDOX developed lung metastases visible as macrometastases at sacrifice and/or micrometastases detectable by HE and vimentin staining of lung sections (Fig 2A and B). Interestingly, tumors from metastatic patients showed high metastatic capacity in mice and indeed 80% of metastatic Ren-PDOX derived from metastatic patients (Fig 2A, graph). Altogether, this suggests that more aggressive and dysplastic tumors maintain the characteristics of malignancy acquired in patients and are more prone to seed and to adapt to different microenvironment for growing.

### Ren-PDOXs reproduce genomic alterations typically found in human ccRCC

Studies from genomic consortiums such as TCGA have demonstrated that ccRCC tumors are characterized by prototypical mutations in some common genes such as VHL, PBRM1, BAP1, SETD2, and TSC1 among others (Atlas, 2013). Thus, we wondered whether Ren-PDOXs molecularly reproduced RCC typical genetic alterations. To this aim, we performed targeted next-generation sequencing (NGS) analysis in DNA extracted from Ren13, Ren28, Ren38, Ren50, Ren86-PDOX tumors using a panel of 397 genes that included all the most common mutated genes in RCC (Appendix Table S3). Collectively, Ren-PDOXs tumors presented 25 different alterations (15 already reported, 10 novel), distributed over 14 genes of Haloplex

panel. We found potentially pathogenic truncating or frameshift mutation in VHL and PBRM1 genes that control processes frequently altered in ccRCC (Atlas, 2013). In particular, three out of five Ren-PDOXs had point alterations in VHL, all already reported in public database, while alterations identified for PBRM1 gene were all novel. Other potentially pathogenic mutations were found in ATM, BAP1, MLL3, mTOR, SETD2, and TSC1 genes (Fig 2D and Appendix Table S4), fully concordant with previously published TCGA mutation profile for clear cell kidney cancer (Atlas, 2013).

Furthermore, in order to determine whether individual PDOX were reproducing the genomic alterations of their original patient's tumor, we performed a whole-exome sequencing (WES) analysis to compare the genomic profile of Ren50-PDOX and its original human tumor specimen. The total number of variants detected was comparable between original human specimen and Ren50-PDOX sample (399 and 355 respectively). Only 9/355 variants were exclusive of PDOX tumor, therefore, 97.5% of Ren-PDOX variants were already present in original human tumor (Fig 2E). Overall, the 408 variants identified by WES were homogeneously distributed over the human karyotype (Fig EV2C). Taken together, these results demonstrate that Ren-PDOXs share genomic characteristics with human tumor reproducing human RCC heterogeneity and complexity, key features for their use as relevant predictive models of human cancer.

### Ren-PDOXs respond to antiangiogenic treatments with varying degrees of efficacy in a patient-specific manner

In order to test antiangiogenic efficacy in our newly developed PDOX models, DC and Beva were administered to mice implanted with four different Ren-PDOX tumors: Ren13, Ren28, Ren50, and Ren86. In all tested Ren-PDOXs, short-term antiangiogenic therapies slowed down tumor growth with tumor weight significantly lower in treated compared to control condition (Fig 3A and B). Indeed, DC and Beva treatments were effective in diminishing vessel number and increasing tumor necrosis (Fig 3C and D) as the causal mechanism of their antitumor effects. Nevertheless, differential effects were observed regarding OS in these treatments: While DC and Beva treatments extended survival in Ren28 and Ren50 tumors, this effect was not observed in Ren13 and Ren86 PDOX models (Fig 3E). These results demonstrate differential responses to antiangiogenics in different PDOX models that recapitulates the differential responses observed in kidney cancer patients treated with antiangiogenics.

### Antiangiogenic treatments promote patient-specific capsular invasion and kidney invasive behavior in Ren-PDOXs

Since pro-invasive effects have been described to be one of the consequences of tumor adaptation to antiangiogenics, we aimed at evaluating tumor invasion at the tumor-kidney interface. Indeed, Ren-PDOX masses grew bound to kidney establishing a tumor-kidney interface at the invasive front. In HE sections, we observed that the interface presented a fibrous capsule. Screening by IHC staining, we found that the fibrous capsule, included renal capsule, was composed by fibronectin (FN) fibers (Fig EV3A). Moreover, we could identify different grade of capsular invasion (CI). Thus, we distinguished tumors whose CI was (i) absent if tumor cells were

separated from normal renal parenchyma by well-defined layers of FN or (ii) present if strands of tumor cells were directly in contact with normal renal parenchyma (Figs 4A and EV3A). In particular,

untreated Ren13 and Ren86 were characterized by low basal CI (around 30% of tumors) whereas 60% of Ren28 and Ren50 tumors presented CI (Fig 4B, control).

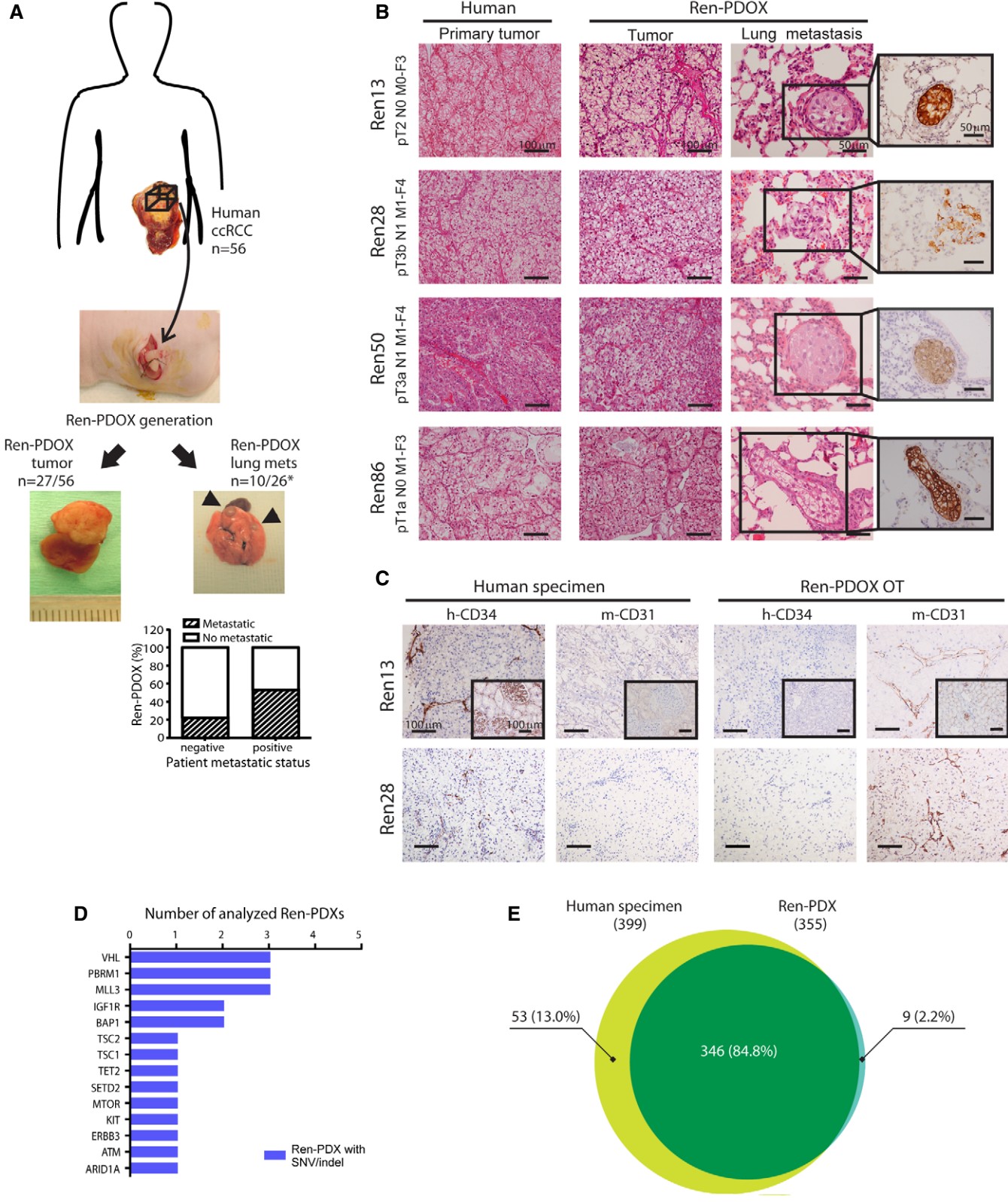

Figure 2.

**Figure 2.  Establishment and histological characterization of Ren-PDOX.**

A   Representation of protocol for the establishment of Ren-PDOXs. Ren-PDOXs were obtained by implantation of a total of 56 ccRCC human specimens on left mouse kidney. Representative images of Ren-PDOX tumor and lung metastases (mets, arrows) are shown. Total number of established and metastatic Ren-PDOXs is reported. Graph describes the proportion of metastatic Ren-PDOXs generated by tumor from patients with negative (pN0 and/or pM0) or positive (pN1 and/or pM1 and/or metachronous) metastatic status. *, one Ren-PDOX has not been evaluated for lung metastasis.

B   Overall mass architecture of paired human specimens and Ren-PDOX tumors observed by HE staining (20X). TNM staging and Fuhrman grade (F) of implanted tumors are reported. For Ren-PDOXs, lung micrometastases in HE and vimentin-stained sections are shown (40X).

C   Human to mouse stroma substitution. CD34/CD31 staining for, respectively, human (h) and mouse (m) blood vessels in original human specimens and paired Ren13 and 28-PDOX are shown (20X). Ren-PDOX OT referred to tumor generated from first implantation of human specimen in mice. Insets show human and mouse kidney as positive/negative controls of staining (20X).

D   Frequency histogram of SNV and/or indels found in Ren-PDOX by high-throughput sequencing in Ren13, Ren28, Ren36, Ren86 (targeted-NGS analysis by Haloplex), and Ren50-PDOX (WES). Only non-synonymous/stop gain SNVs and frameshift indels novel or with MAF ≤ 0.01 are included.

E   Genetic comparison of original human specimen (patient #50) and its derived PDOX (Ren50-PDOX) by WES. Venn diagram shows number and percentage of variants called in human specimen (yellow), in its PDOX (blue) and overlapped variants (green).

Since it has been recently suggested that antiangiogenic therapy could modify RCC tumor-normal kidney parenchymal interface (Kawakami *et al*, 2016), we wondered if treatment could exert comparable effects on invasive front of Ren-PDOX tumors. When the four PDOXs were treated with DC and Beva, both treatments exerted differential effects on tumor invasive capacity in each PDOX model. In Ren13 and Ren86, DC and Beva treatment produced significantly more CI than controls (Fig 4B) resulting in >2-fold higher local invasion (Figs 4C and EV3B and C). In contrast, in Ren28 and Ren50 the percentage of tumor masses showing CI was similar in control and treated animals (Fig 4B), therefore, VEGF/R blockade did not modify tumor invasive behavior of this other subset of tumors (Figs 4C and EV3C). Therefore, our findings show that VEGF-targeted therapy impaired tumor growth in all PDOX tested, but only increased the invasive capacity of a specific subset of ccRCC tumors, indicating a patient-specific pro-invasive response.

In order to test whether these pro-invasive effects were also observed when a clinically approved antiangiogenic was administered to our Ren-PDOX, we treated a pro-invasive (Ren13BM) and a non-pro-invasive (Ren28) models with sunitinib and evaluated invasion. As shown in Fig 4D, sunitinib treatment increased both CI (bar graphs) and tumor local invasion (box plots) in Ren13BM but not in Ren28. Furthermore, sunitinib pro-invasive analysis was extended to a new set of five PDOXs that also showed patient-specific differences in capsular and local tumor invasion. In this set, three out of five PDOXs show enhanced tumor invasion (Fig EV3D). Thus, this suggests that sunitinib also produces increased invasive capacity of a specific subset of ccRCC tumors, indicating a prominent patient specificity of this pro-invasive response.

**Antiangiogenics promote patient-specific variable responses in systemic dissemination and metastasis**

We then wondered whether contrasting effects on local invasion could have different consequences on development of metastasis in Ren-PDOXs. The use of spontaneous metastatic models to assess change in metastatic dissemination is challenging. Thus, we compared the effects of VEGF/R inhibition on two independent Ren-PDOXs showing high basal metastatic capacity but opposite invasive behavior upon treatments (Ren13BM and Ren28). Interestingly, we found that in Ren13BM the pro-invasive effects of VEGFR-targeted therapies on local invasion was associated with augmentation of

lung metastasis (Fig 5A) and indeed incidence of metastasis as well as number of metastatic foci significantly increased in DC-treated compared to control mice (Fig 5B and C, left). We also observed higher metastatic dissemination in Beva-treated mice, although not significantly (Fig 5B and C, left). On the other hand, in Ren28, antiangiogenic treatments did not modify incidence of metastases in treated mice compared to controls, nor number of metastatic foci per mouse (Fig 5B and C, right), in accordance with unmodified local invasion.

Taken together, our observations suggest that inhibition of VEGF signaling pathway could have different inter-tumor impact therefore resulting in contrasting local and subsequently systemic effects (Fig 5D).

**Tumor cell-specific expression profile defines molecular hallmarks of pro-invasive behavior**

In order to get a mechanistic insight into the molecular basis of tumor predisposition to acquire an invasive behavior upon antiangiogenic therapies, we performed a novel RNA sequencing (RNA-seq) technique that allows for discrimination of tumor and stromal expression profiles. Briefly, whole-genome RNA-seq of a pro-invasive PDOX model (Ren13) and a non-pro-invasive PDOX (Ren50) in basal untreated condition was performed where human and murine transcripts can be unequivocally distinguished, allowing, respectively, the analysis of tumoral and stromal contribution to gene expression profiles (Fig 6A). Results from tumor-derived (human origin) comparative expression showed that most genes were present in both tumor types (85.1%, Fig 6A). However, a fraction of the transcripts was expressed exclusively either in Ren13 (6.7%) or in Ren50 tumors (8.2%, Fig 6A). As expected, stromal (murine origin) gene expression overlap between Ren13 and Ren50 was even higher (94.7%, Fig 6A), even though a small proportion of stromal genes was found specific for each tumor type (1.6% and 3.7% for Ren13 and Ren50, respectively, Fig 6A). These results demonstrate the intrinsic genetic differences from these two Ren-PDOX tumors mainly due to tumoral contribution.

To identify a signature of molecular pathways differentiating divergent invasive behavior upon antiangiogenics, gene set enrichment analysis (GSEA) was performed from basal untreated RNA-seq expression data of the tumor cell component (human origin) (Fig 6B). GSEA analysis identified UV response down, KRAS signaling up, coagulation, angiogenesis, and Wnt/β-catenin signaling as the most enriched pathways in pro-invasive tumors (Fig 6B and

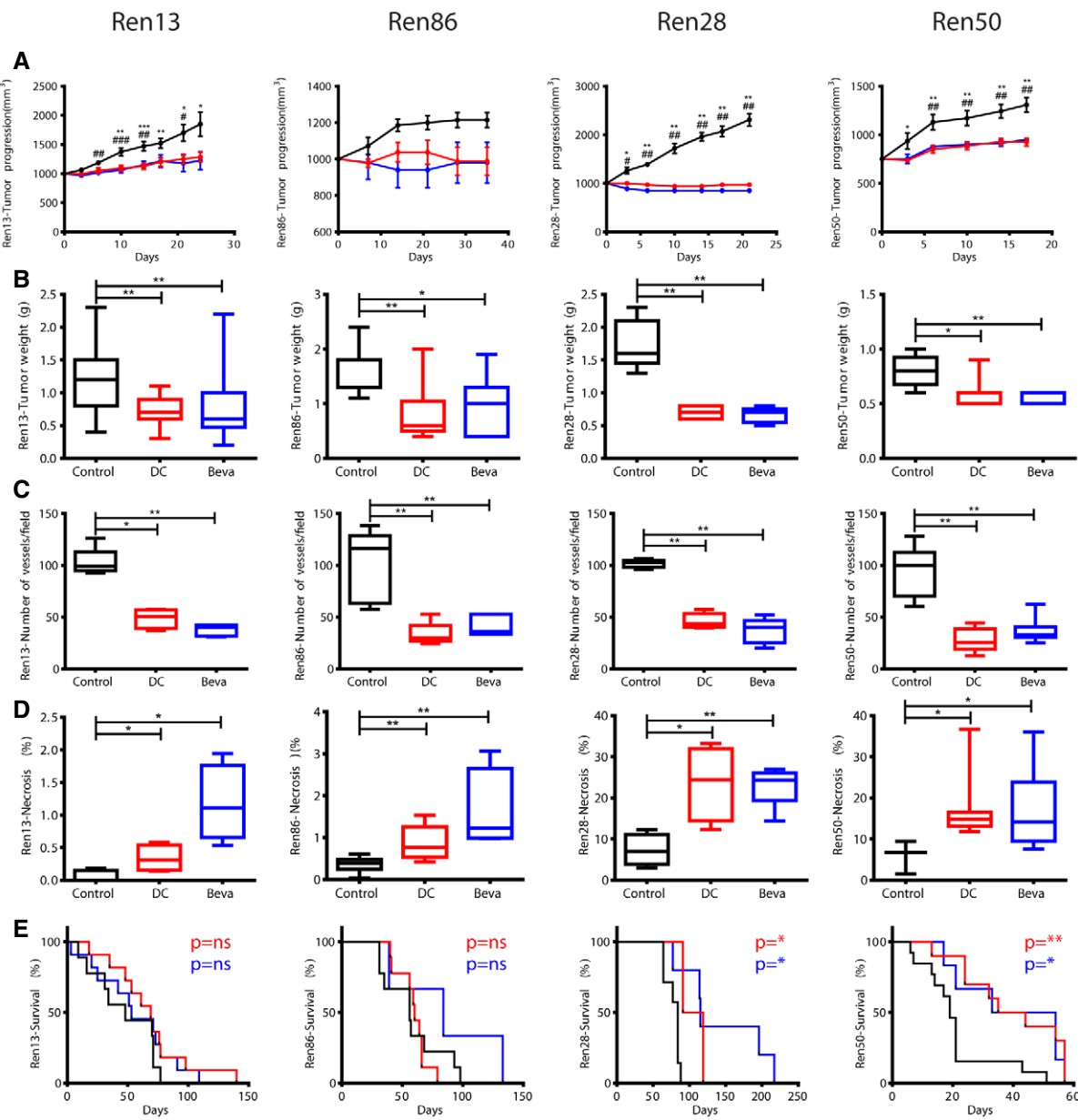

**Figure 3. Robust antitumor effects of anti-VEGF/R therapies in Ren-PDOX.**

A–E  Effects of DC and Beva treatments on Ren13, Ren86, Ren28, and Ren50-PDOX. Evaluation of tumor progression average and SD (A) in control (black) compared to DC (red, \*P < 0.05) and Beva (blue, #P < 0.05) treatments from five to 13 animals per tumor and treatment group by Mann–Whitney test. Quantification of tumor weight (B), vessel number (C) and tumor necrosis (D) in control and treated tumors from four to five samples/tumor/treatment group by Mann–Whitney test. Box plots indicate median, Q1/Q3 and max/min value whiskers. Evaluation of overall survival (E) comparing Control (black), DC101 (red), and Beva (blue)-treated mice from four to 13 mice/tumor/treatment group by Mantel-Cox test where \*P < 0.05, \*\*P < 0.01, and \*\*\*P < 0.001.

Appendix Fig S1A). On the other hand, non-pro-invasive tumors were enriched in gene sets involving E2F targets, G2/M checkpoint, estrogen response late, epithelial–mesenchymal transition (EMT), KRAS signaling down, reactive oxygen species (ROS) pathway, and IL2/STAT5 signaling (Fig 6B and Appendix Fig S1A). Remarkably, pro-invasive tumors showed increased KRAS signaling in comparison to non-pro-invasive (more expression of genes upregulated by KRAS and less of genes downregulated by KRAS), suggesting that this pathway may be an important axis implicated in the tumor

predisposition to acquire a more malignant phenotype after antiangiogenic therapy.

In order to filter the KRAS signature genes to obtain a smaller more specific geneset, an extensive series of clear cell kidney cancer patients from TCGA was used (Atlas, 2013). The possible association between candidate normalized (z-score) gene expression and clinical variables in primary TCGA-KIRC tumor samples was studied by age- and gender-adjusted linear regressions (Appendix Fig S1B). Guided by both Tumor Stage (S), Tumor Size (T), and Metastasis

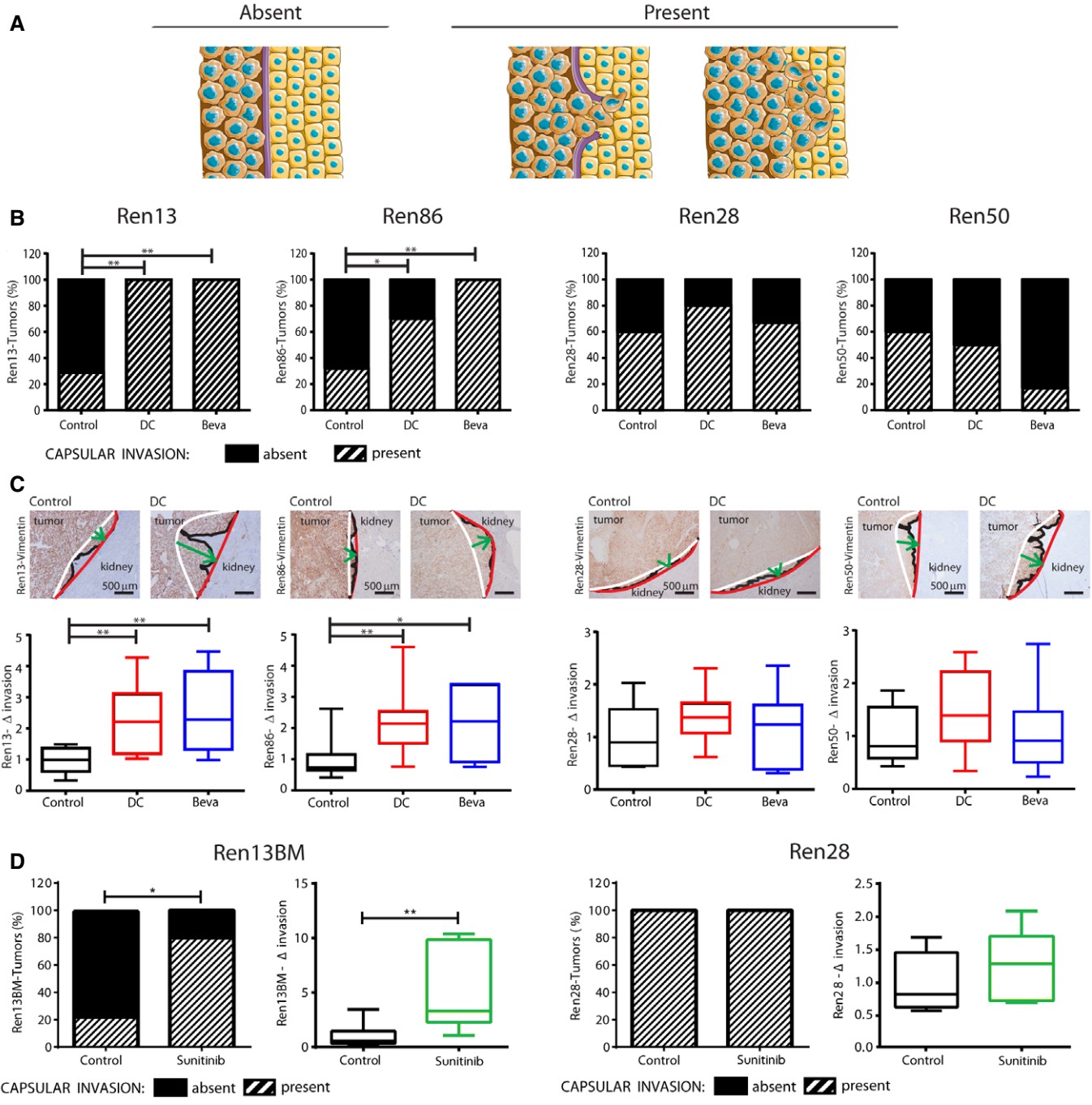

**Figure 4. Effects of anti-VEGF/R therapies on local invasion.**

A   Capsular invasion was defined as (i) absent if tumor cells were separated from normal renal parenchyma by well-defined layers of FN or (ii) present if strands of tumor cells were directly in contact with normal renal parenchyma

B, C   Effects of DC and Beva treatment on Ren13, Ren86, Ren28, and Ren50-PDOX on capsular invasion (B) and tumor front of invasion (C). (B) Incidence of Capsular invasion (stripes) in each model and treatment. 5–8 animals per model and treatment group were analyzed by Chi-square test where *$P < 0.05$, **$P < 0.01$. (C) Representative images of Vimentin staining (4X), as described in Fig 1, and box plot quantification of fold-invasion (Δ invasion) of treated *vs* control tumors indicating median, Q1/Q3 and max/min value whiskers. 6–12 animals per model and treatment group were analyzed by Mann–Whitney test where *$P < 0.05$, **$P < 0.01$.

D   Effects of sunitinib treatment on Ren13BM and Ren28 on capsular invasion and tumor front invasion. Bar graphs show incidence of capsular invasion (stripes) in control and sunitinib-treated tumors. 3–9 animals per model and treatment group were analyzed by Chi-square test where *$P < 0.05$, **$P < 0.01$. Box plots represent median, Q1/Q3 and max/min value whiskers of fold-invasion (Δ invasion) of sunitinib-treated vs control tumors. 5–8 animals per model and treatment group were analyzed by Mann–Whitney test. *$P < 0.05$, **$P < 0.01$.

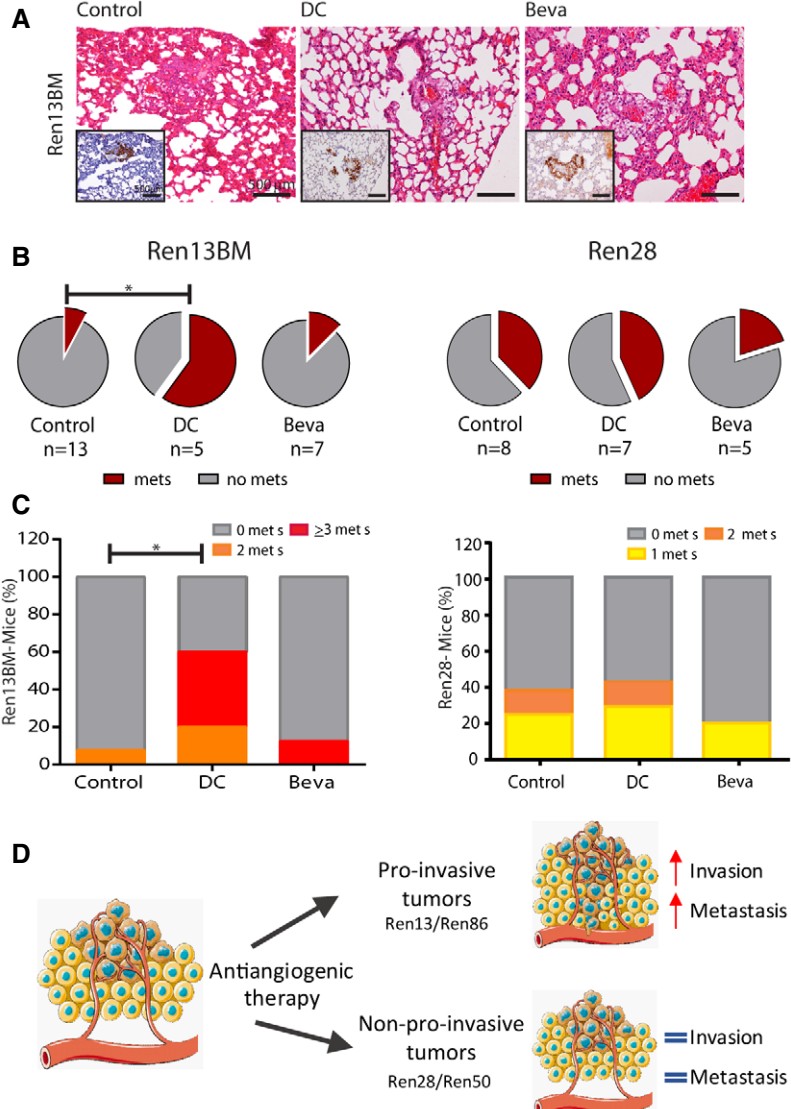

**Figure 5. Effects of anti-VEGF/R therapies on metastasis.**

A–C   Evaluation of lung micrometastases using HE and vimentin-stained lung sections upon antiangiogenic therapies in Ren13BM and Ren28. (A) Representative images
of lung metastasis in Ren13BM upon antiangiogenic therapies. (B) Incidence (pie chart, *$P < 0.05$ by Chi-square) and proportion of mice with 0, 1, 2, or ≥3
metastases (mets) per lungs (C) (*$P < 0.05$ by Mann–Whitney test) are shown.

D   Anti-VEGF/R treatments exacerbate the invasive capacity quantified as capsular invasion in the tumor-kidney interface in some Ren-PDOXs. Moreover, the
evaluation of lung metastasis confirms the pro-malignant effects of antiangiogenics in some Ren-PDOXs, recapitulating the inter-patient variable response.

(M), only 19 genes were found differentially expressed (FDR-adjusted P-value < 0.05) in the more advanced/aggressive subset of all these clinical characteristics (Fig 6C).

Validation of filtered candidates using a unique series of 39 RCC patients whose microvascular invasion and renal vein involvement were specifically annotated. Candidate signature genes differential expression analysis was performed in the GSE29609 patient series as a validation dataset, evaluating renal vein involvement and microvascular invasion. As shown in Figs 6D and EV4, only two of the 19 selected candidate genes, namely ALDH1A3 and MAP7, were robustly differentially expressed in tumors with microvascular invasion or with renal vein involvement.

In order to define the prediction capacity of our two candidate genes, we determined them in both pro-invasive and non-pro-invasive Ren-PDOXs and found that ALDH1A3 clearly associates with pro-invasiveness at protein and mRNA levels, but MAP7 did not (Fig 7A and B and EV5A). To extend and confirm these data, the two candidates were evaluated in sunitinib-treated Ren-PDOXs. Results confirmed that ALDH1A3 was significantly associated with pro-invasive phenotype in these samples (Fig 7C) and its protein levels significantly correlated to invasion (Fig 7D), while MAP7 did not (Fig EV5B–D).

In order to test the predictive clinical value of ALDH1A3, we evaluated a series of 15 patients clinically treated with sunitinib

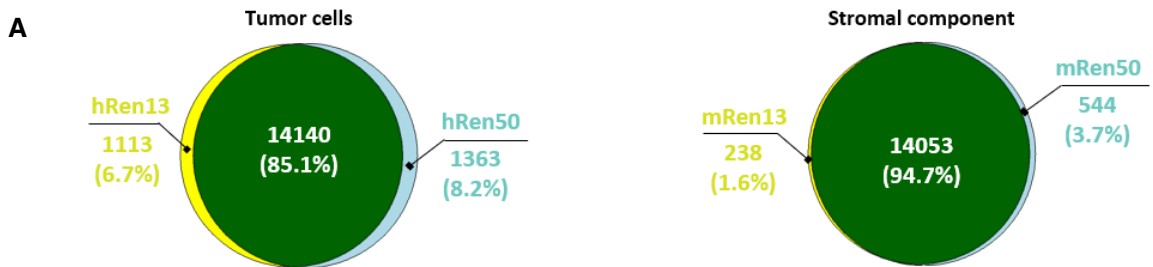

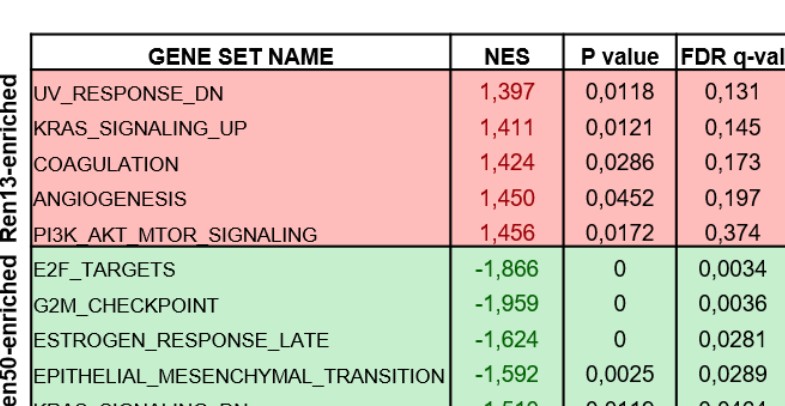

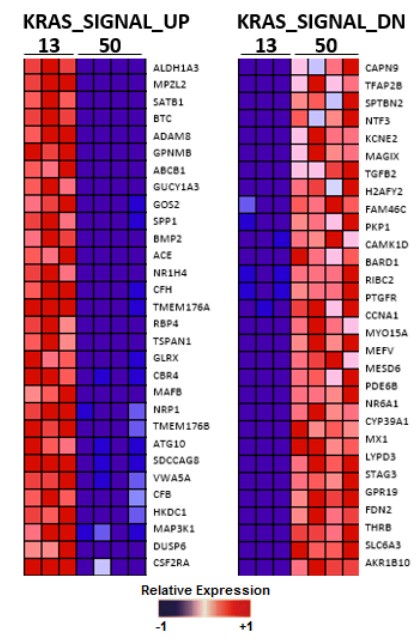

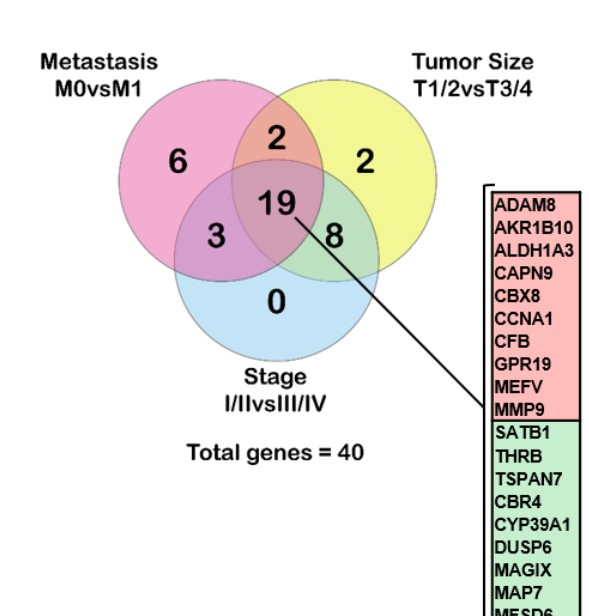

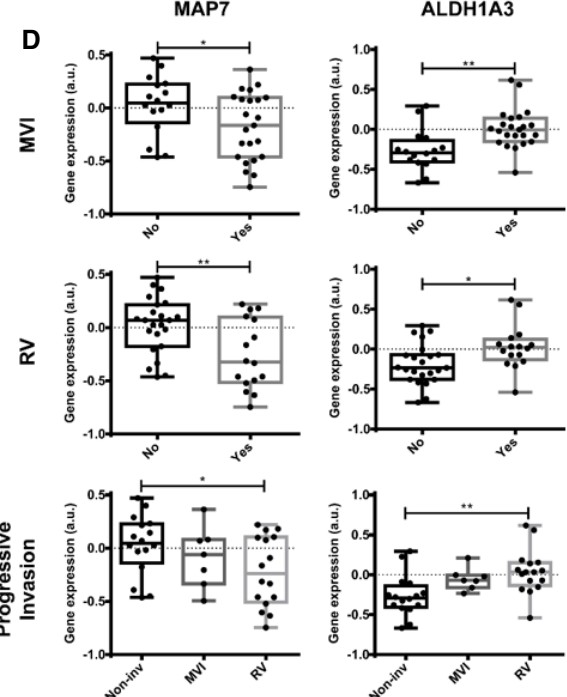

Figure 6.

**Figure 6. Segregation of tumor and stromal expression profile to define tumor-specific molecular hallmarks of pro-invasive behavior.**

A  Comparison of tumor-derived (human, left) and stroma-derived (mouse, right) differentially expressed genes in a pro-invasive (Ren13) and non-pro-invasive (Ren50) Ren-PDOX tumor models, obtained from species-specific RNA-seq analysis. Venn diagrams illustrate gene expression differentials and overlap between Ren13 and Ren50 tumor types.

B  Tumor specific (human) enriched gene sets in Ren13 and Ren50 Ren-PDOX tumors, after GSEA analysis from RNA-seq data. Gene sets most significantly enriched in Ren13 or Ren50 are indicated in the table (left), including their normalized enrichment score (NES), P-value, and false discovery rate (FDR q-value). Heatmaps illustrate expression levels of the genes responsible for the enrichment of KRAS signaling up and down in Ren13 and Ren50, respectively (right).

C  Venn diagram comparing differentially expressed genes in the KRAS signature among Stage (blue), T (yellow), and M (red) from the TCGA-KIRC primary tumors cohort. Only significant (FDR-adjusted P-value < 0.05) age- and gender-corrected correlated genes are included. 19 genes are listed separating upregulated (red) and downregulated (green) in more advanced/aggressive tumors.

D  Validation of candidate gene expression in unique series of 39 RCC patients (GSE29609), where microvascular invasion (MVI) and renal vein involvement (RV) are specifically annotated. Progressive invasion relates to the three progressive degrees of invasion toward the vena cava in this series: 1st, no invasion; 2nd, microvascular invasion; and 3rd, renal vein involvement. The gene expression is shown for each variable and condition in box plots representing median, Q1/Q3 and max/min value whiskers, *P < 0.05, **P < 0.01 by Mann–Whitney test.

whose pro-aggressive response after therapy was fully annotated. As shown in Fig 7E, patient's response to therapy was categorized as disease progression with new metastasis/ overt local infiltration, or disease progression with no new lesions/ local progression. Tumor specimen analysis before treatment in this series confirmed that ALDH1A3 can significantly discriminate pro-aggressive response in patients, showing not only association but also a linear trend of ALDH1A3 levels to pro-aggressive response (Fig 7F). Furthermore, the predictive power of ALDH1A3 was evaluated using a ROC curve study with very significant results and defining its initial sensitivity and specificity (Fig 7G).

Overall, our data clearly defines ALDH1A3 as a possible predictive factor of pro-aggressiveness and could be used to predetermine tumor predisposition to acquire aggressive/invasive capacity in RCC patients treated with antiangiogenic therapies (Fig 7H).

## Discussion

Increased tumor invasiveness and dissemination have been described as possible mechanisms of evasion from VEGF-targeted therapies in several preclinical and clinical studies (Ebos & Kerbel, 2011; Kuczynski et al, 2016). Here, by developing a significant series of ccRCC patient-derived orthotopic PDOX mouse models, we sought to investigate the consequence of VEGF pathway inhibition on RCC invasiveness and metastatic potential and to identify new biomarkers to predict the aggressive capacity of tumors.

Cell-based xenografts had been classically used in preclinical research, anticipating the generation of patient-derived xenografts that better fit with the development of personalized medicine (Tentler et al, 2012). Although showing predictive limitations, they are easy to manage and fast and homogenously growing, therefore, representing a useful tool for validation of parameters to measure biologic evolution of tumors upon therapy. Hence, we used 786-O-derived tumor piece orthoxenograft to set up appropriate parameters to evaluate tumor local invasion and spontaneous metastatic dissemination. Using this tool, we firstly observed that in RCC the inhibition of VEGF/R signaling, although affecting tumor growth, rapidly resulted in augmented local tumor invasiveness and increased metastatic dissemination, in accordance with effects reported for other cancer models (Paez-Ribes et al, 2009; Lu et al, 2012; Sennino et al, 2012).

To move to a closer bedside perspective, we developed a series of tumor piece-derived orthotopic RCC-PDOX models. Engraftment of implanted tumor pieces was around 50%. Interestingly, tumors from metastatic patients exhibited very high engraftment rate and

metastatic capacity. Moreover, they mimic human RCC progression from primary tumor growth to development of metastases, thus confirming and strengthening previous published observations (Sivanand et al, 2012; Thong et al, 2014). Interestingly, a representation of Ren-PDOX collection showed potentially pathogenic mutations in most common mutated genes in RCC such as VHL, PBRM1, BAP1, SETD2 (TCGA, 2013; Brugarolas, 2014; Liao et al, 2015). Moreover, we and others found that PDOX genomic profile overlapped with paired tumor from patients (Hidalgo et al, 2014). Hence, they reproduce patient variability and could be considered appropriate preclinical tools to identify the crucial aspects of neoplastic evolution and to assess tumor response to therapy.

In patients, most of the renal cancers grow surrounded by a fibrous pseudocapsule in between tumor and normal renal tissue. Capsular invasion with tumor protruding into the pseudocapsule has been described and CI has been identified as possible marker of tumor aggressiveness and perhaps metastatic potential (Cho et al, 2009; Snarskis et al, 2016; Volpe et al, 2016). Patients with organ-confined or locally advanced RCC tumors could be candidates for neoadjuvant therapies to potentially target primary mass and prevent local and metastatic disease. However, preoperative therapies still lack of considerable consensus and possible consequences on tumor progression have been suggested (Griffioen et al, 2012; Borregales et al, 2016). To note, in a recent study, Kawakami et al evaluated the effects on the histology of tumor-kidney interface of the antiangiogenic axitinib administered as neoadjuvant therapy for ccRCC. They found discontinuous pseudocapsule in the majority of axitinib-treated cases and higher capsular invasion in axitinib compared to controls, although not significant probably due to the small case numbers (Kawakami et al, 2016).

Tumor-stroma interaction and physical structural support of non-transformed cells may profoundly influence the effects of therapy on tumors (Junttila & de Sauvage, 2013). RCC-PDOXs generated from pieces of tumor maintain histological architecture and structure of original human specimens and are therefore suitable predictive tools to study the impact of therapy on tumor growth and evolution (Karam et al, 2011; Sivanand et al, 2012; Hidalgo et al, 2014; Pavia-Jimenez et al, 2014; Thong et al, 2014). Interestingly, we found that histologic sections of our Ren-PDOXs showed a fibrous capsule at tumor-kidney interface, physically resembling the boundary defined by pseudocapsule in human tumor. Independent Ren-PDOXs were characterized by different capacity to invade the capsule once again reproducing the human inter-tumor heterogeneity. Intriguingly, we showed that the inhibition of VEGF pathway could have an impact on tumor margins and invasiveness. Furthermore, the results of an ambitious but

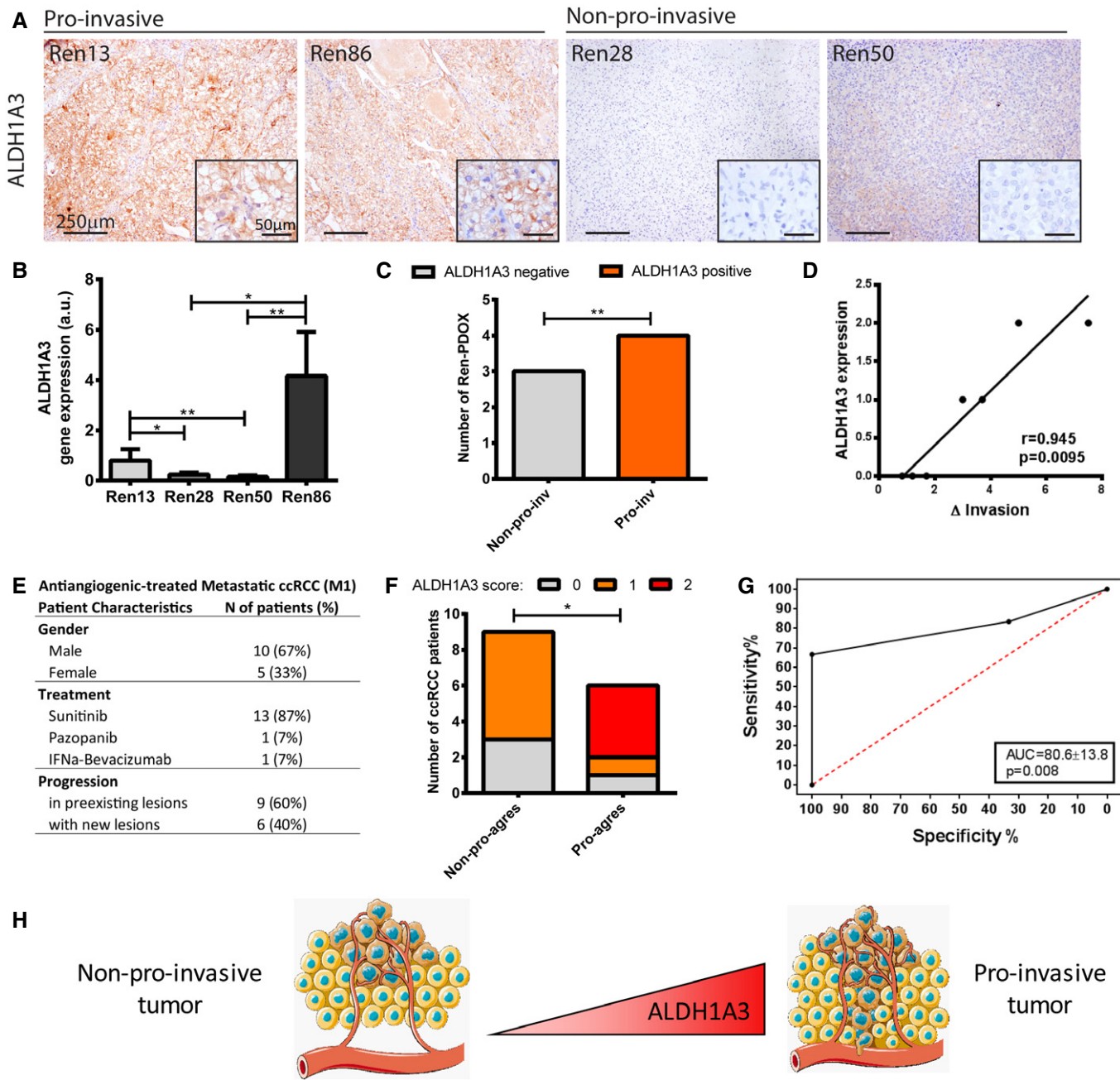

**Figure 7. Validation of ALDH1A3 as a tumor-specific biomarker of pro-invasive behavior.**

A    Representative images of ALDH1A3 expression on pro-invasive (Ren13 and Ren86) and non-pro-invasive (Ren28 and Ren50) tumors.

B    Confirmation of ALDH1A3 mRNA levels of gene expression in pro-invasive (Ren13 and Ren86) and non-pro-invasive (Ren28 and Ren50) tumors. Average ± SD of 4–6 independent replicates per group were analyzed by Mann–Whitney test *$P < 0.05$ **$P < 0.01$.

C, D  Association of pre-treatment ALDH1A3 protein levels by IHC on a series of Ren-PDOX models treated with sunitinib. (C) Representation of ALDH1A3 expression on pro-invasive and non-pro-invasive Ren-PDOX after sunitinib treatment (**$P < 0.01$ by Chi-Square test). (D) Correlation of pre-treatment ALDH1A3 protein levels by IHC with increased tumor invasion found after sunitinib treatment. $n = 7$, Spearman's non-parametric correlation, $P = 0.0095$.

E    Patient characteristics of a series of 15 metastatic ccRCC patients specifically annotated for their type of progression after antiangiogenic therapy. 60% progressed in preexisting lesions (non-pro-aggressive) while 40% progressed with new lesions (pro-aggressive).

F    Representation of ALDH1A3 score (0 = absent, 1 = low, 2 = high expression) on pre-treatment samples of pro-aggressive and non-pro-aggressive ccRCC patients described above. $n = 15$, Chi-square test for independence $P = 0.015$, and Chi-square test for trend $P = 0.030$ (*).

G    ROC curve for the ALDH1A3 score on 15 ccRCC patients. ROC variation method was used due to categorical predictor factor. AUC = 80.6 ± 13.8, Sensitivity 83.3% [95%CI: 35.9–99.5], Specificity 33.3% [95%CI: 7.5–70.0], empirical $P = 0.008$ by Bootstrapping method ($n = 1,000$).

H    Levels of ALDH1A3 differentiate between non-pro-invasive and pro-invasive tumors. High levels of ALDH1A3 in samples before treatment are associated with more invasive tumors after antiangiogenic treatment.

Source data are available online for this figure.

relevant experiment on spontaneous metastatic PDOXs, although possibly limited by sample size, showed that increased local invasion could evolve in higher metastatic dissemination.

Our hypothesis is that inhibition of VEGF signaling may modify the evolution of encapsulated or less locally infiltrative tumors switching on their invasive behavior and eventually leading to enhanced metastatic dissemination. On the other hand, treatments do not affect local and systemic dissemination of tumors characterized by marked basal invasion. Interestingly, recent papers suggest that tumors with infiltrating cells growing directly in contact with normal tissue parenchyma are less dependent on sprouting angiogenesis and therefore less sensitive to angiogenesis inhibition (Frentzas *et al*, 2016; Kuczynski *et al*, 2016).

Altogether, our findings suggest that in specific RCC patients antiangiogenics could change the natural progression of neoplasia and promote a tumor invasive behavior. Therefore, controversy about the pro-invasive effects of antiangiogenics probably reflects the variability of tumor response to therapy and should possibly be solved taking into account inter-tumor heterogeneity. More importantly, in the clinics sunitinib treatment in RCC patients produces a well-documented increased PFS that is not always resulting in expanded OS (Rini *et al*, 2008; Motzer *et al*, 2009). Our data seem to indicate that this apparent discrepancy could be related to a therapy direct antitumor benefit in all PDOXs tested, but a differential benefit in OS in pro-invasive or non-pro-invasive tumors. In detail, OS benefit is achieved in two non-pro-invasive PDOXs, while in the two pro-invasive ones there is no benefit in OS. Therefore, our data suggest that antiangiogenics could be producing a PFS benefit in a majority of patients, but the pro-invasive response in some of them could erode this benefit into no OS benefit.

One potential limitation of our study is the possibility that the increase in metastasis caused by antiangiogenics in certain models reflects the fact that the treated mice survived for a longer period of time and, as such, this may increase the probability of some mice eventually developing metastatic disease. Nevertheless, in our metastatic models, the increase in metastasis after antiangiogenics (Fig 5C, left) happened in animals that survived similarly to controls (Fig 3E, first panel), while in other models metastasis was not changed (Fig 5C, right) but animals survived longer upon treatment (Fig 3E, third panel).

This patient-specific differential response to antiangiogenics calls for new efforts to identify specific biomarkers and tumor features for appropriate selection of patient to be treated. Analysis of RNA-seq data comparing untreated gene expression profiles of pro-invasive and non-pro-invasive tumors suggests that intrinsic features of individual tumors could account for different invasive/metastatic behavior, and pinpoints a KRAS signature as the main difference between them, consistent with findings in other tumor types (Poulin & Haigis, 2017). Furthermore, TCGA filtering of this signature and subsequent validation in an invasion-annotated patient series revealed two biomarkers that define tumor aggressiveness. Therefore, our data open new avenues to test these two biomarkers to predict the response to antiangiogenic treatment in terms of tumor invasiveness and dissemination in ccRCC patients. Furthermore, these findings may lead to the identification of novel targets and therapeutic strategies able to suppress the undesired pro-malignant effects of antiangiogenics. PDOXs completely recapitulate different aspects of human tumor, including relation between tumor and surrounding normal tissues as well as heterogeneous and specific response to therapy. Therefore, they could represent a fundamental tool for preclinical studies in the era of personalized medicine maintaining the complexity of human tumors and diversity of cancer patients.

# Materials and Methods

### Generation of 786-O orthoxenografts

For the generation of first 786-O cell-derived tumor, 5-week-old to 6-week-old male nude mice provided by different vendors (Charles River, Envigo...) were anesthetized, left kidney exteriorized and $1 \times 10^6$ cells were directly injected into kidney capsule. 7860- cell line was provided by the original donor laboratory (B. Jiménez, IIB, CSIC-UAM, Madrid Spain) and tested for micoplasma contamination at a weekly biweekly frequency. Once tumor was palpable (approximately 1,000 mm$^3$ of volume) animals were sacrificed and a piece of tumor of 3–5 mm was implanted in the kidney of a new recipient mouse. Tumors were perpetuated *in vivo* by consecutive passages into new recipient nude mice. All experiments were developed at our AAALAC-accredited facility (#1155) according to our Institute's Animal Research Committee acceptance, and following Spanish laws and European directives on ethical usage of rodents for animal research (local government GenCat DTS approval Project# 9727, Procedure#4899).

### Generation of Ren-PDOX from human specimens

ccRCC patient-derived orthoxenograft models (Ren-PDOX/Ren) were generated by orthotopic implantation in nude mouse kidney of fresh human specimens of ccRCC obtained at surgical resection or biopsy from the Bellvitge or Vall d'Hebron Hospitals (Barcelona, Spain) under local ethics committee's approved protocols (CEIC approvals ref. PR322/11 and PR[AG]240/2013). Patients enrolled provided an informed consent to participate in the study, which complied with Spanish laws and European directives and are aligned with the WMA Declaration of Helsinki and the Department of Health and Human Services Belmont Report. 56 ccRCC specimens were collected at surgery and immediately implanted in mice left kidney and took a median of 53 days CI95% [40–95 days] to grow to 1,000 mm$^3$. Ren-PDOX were then maintained and perpetuated *in vivo*, as previously described. Tumor volume was followed up by palpation in live animals using a scale developed by reference RMN images and luciferase experiments, and confirmed at sacrifice by tumor weight and volume. Clinical details of patients and characteristics of tumors at implantation are available in Appendix Tables S1 and S2. For Ren13, we generated Ren13-PDOX and the paired metastatic variant Ren13BM-PDOX implanting in kidney of mice, respectively, piece of the human primary ccRCC and metachronous piece of brain metastasis from the same patient.

### Treatment schedule

The following antiangiogenic regimens were used: (i) anti-mouse VEGFR2 blocking antibody (DC101) intraperitoneally (i.p.) administered, 1 mg/dose/mouse, twice a week. DC101 was collected from a

hybridoma culture (ATCC, Manassas, USA); (ii) anti-human VEGF monoclonal antibody (Bevacizumab, Avastin, 25 mg/ml, Roche Pharma AG, Grenzach-Wyhlen, Germany), i.p., 5 mg/kg/dose, twice a week. The administration started when tumor was palpable. To study tumor local invasion, the duration of treatment was calculated based on the median of control lifespan. To study metastasis, therapies were administered until survival of each mouse. End-point survival was defined by animal health welfare criteria (scale of diverse events of signs and symptoms of discomfort and pain) and was reached due to primary tumor mass growth and infiltration.

## Immunohistochemistry

Three-micrometer thick of paraffin-embedded tumor or lung tissue sections were rehydrated and processed by standard procedure for HE staining and immunohistochemistry (IHC). The following primary antibodies were used: mouse anti-vimentin 1/200 (clone V9, Thermo Fisher Scientific, Inc., Cambridge, UK); mouse anti-CD34 1/100 (clone QBEND-10, ab8536, Abcam, Cambrige, UK); rabbit anti-CD31 1/50 (ab28364, Abcam); rabbit anti-fibronectin 1/150 (ab2413, Abcam); rabbit anti-ALDH1A3 1/100 (ab129815); rabbit anti-MAP7 1/100 (NBP1-84853). The EnVision system of labeled polymer-HRP anti-mouse or anti-rabbit IgG (DakoCytomation, Agilent Technologies, Santa Clara, USA) was used undiluted as a secondary antibody. IHC of patients' tumor samples was scored 0, 1, or 2 to define absent, low or high expression of ALDH1A3 and MAP7, respectively.

## Evaluation of tumor local invasiveness

Evaluation of tumor local invasion was done by ImageJ software tracing the tumor-kidney interface in 4X images of vimentin-stained sections taken through all invasive front and evaluating widest extension of tumor protrusion into kidney parenchyma for each image (depth) (Fig EV1B). Then, average of depth was calculated for each tumor. Due to the positive correlation found between depth and tumor weight in untreated tumors (Fig EV1C), we calculated the invasion as depth (μm) normalized for tumor weight (g). To compare different experiments, fold-invasion (Δ invasion) was calculated as invasion normalized to the average of invasion of control group of each experiment.

## High-throughput DNA sequencing

For high-throughput sequencing, DNA was extract from snap frozen Ren13, Ren28, Ren38, Ren50, and Ren86-PDOX tumor samples using the DNeasy kit (Qiagen, Hilden, Germany) according to manufacturer's instructions. Next-generation sequencing (NGS) was performed at Cancer genomics core Lab at VHIO (Barcelona, Spain). Whole-exome sequencing (WES) was performed in Hiseq 2500 sequencer (Illumina, Inc., San Diego, USA). Somatic variants were called when supported by at least seven reads, representing at least 7.5% of the total reads. Targeted-NGS was performed in MiSeq sequencer (Illumina, Inc., San Diego, USA) using a custom Haloplex panel (Agilent Technologies, Santa Clara, USA) of 397 genes (Appendix Table S3). Variants were retained with a threshold of depth ≥ 200 and variant allele frequency ≥ 10%. SNVs and indels characterization were performed using the following public databases: Genome Browser, COSMIC, VARSOME, and ICGC Data Portal. Only non-synonymous/stop gain SNVs and frameshift indel novels or with reported allele frequency ≤ 1% (MAF ≤ 0.01) were included in Fig 2D and E and Appendix Table S4.

## Massive RNA sequencing

The RNA extraction of tumor specimens was performed using the RNeasy Mini Kit (Qiagen, Hilden, Germany), according to the manufacturer's recommendations. Samples were sequenced at Centro Nacional de Análisis Genómico (CNAG-CRG, Barcelona, Spain). A modified TruSeq™ Stranded Total RNA kit protocol (Illumina Inc.) was used to prepare the RNA-seq libraries from samples. Each library was sequenced using TruSeq SBS Kit v3-HS (Illumina), in paired-end mode with a read length of 2 × 76 bp. Image analysis, base calling, and base quality scoring of the run were processed by integrated primary analysis software—Real Time Analysis (RTA 1.13.48) and followed by generation of FASTQ sequence files by CASAVA 1.8.

RNA-seq reads were aligned to the human (GRCh38/hg38) and mouse (GRCm38/mm10) reference genomes using STAR (version 2.5.1b) and GSNAP (version 2015-06-23), respectively, with ENCODE parameters for long RNA. Genes were quantified using RSEM (version 1.2.28) and read counts were used as input for DESeq2 (version 1.10.1). The cut-off for considering a gene significantly up-sampled or down-sampled was FDR < 5%. Subsequently, Gene Set Enrichment Analysis (GSEA) was used to detect coordinated expression within samples using default parameters.

## Patient data

Clinical validation of gene expression with clinical variables was performed from two series of ccRCC patients: TCGA-KIRC (Atlas, 2013) and GSE29609 (Edeline et al, 2012). (i) In TCGA study, 528 KIRC patients were analyzed by age- and gender-adjusted linear regressions comparing normalized (z-score) gene expression to clinical variables as Tumor Stage (I/II = 318, III/IV = 205), Tumor Size (T) (1/2 = 338, 3/4 = 190), Lymph nodes Invasion (N) (0 = 254, 1 = 238), and Metastasis (M) (0 = 416, 1 = 78); to obtain their fold change (computed using foldchange method at gtools R library), gene expression beta regression coefficients and nominal and FDR-adjusted $P$ values. (ii) In the GSE29609 study, 39 ccRCC patients were evaluated and normalized log10 ratio Cy5/Cy3 gene expression values were analyzed in patients with and without Renal Vein Involvement (RV; 0 = 23, 1 = 16) or Microvascular Invasion (MVI; 0 = 16, 1 = 23) by Unpaired Fisher's exact test.

Series of patients for prediction of response were gathered in a clinical study approved by local ethics committee (ref. HUVH-PR (AG)240/2013). Fifteen patients were included and gender, treatment regimen and type of progression information were gathered for this study, together with tumor specimen from either nephrectomy or tumorectomy prior to treatment. Clinical data and samples were handled following ethical procedures under local ethics committee approval.

## Statistical analysis and data representation

Results are presented as box plots (min to max values and median). Depth and tumor weight correlation were assessed by Spearman

### The paper explained

**Problem**

Resistance to anti-cancer therapies is a clinically relevant event, and the consequences of antiangiogenic therapies on tumor invasiveness and metastasis are still an open debate, particularly for kidney cancer patients for whom these treatments are first-line standard of care.

**Results**

To study this clinical problem, we have developed an extensive series of kidney cancer mouse models derived from 27 patients that maintain the genetic, cellular, morphologic, and dissemination properties of each of those patients, thus representing the heterogeneity of this patient population.

These unique models revealed the patient-specific pro-malignant effects of antiangiogenics: antiangiogenic therapies resulted in increased invasiveness and metastatic dissemination in a subset of tumors, while in others, aggressiveness remained unchanged. By tumor-specific RNA sequencing and further validation in several series of patient samples, we were able to pinpoint a putative discriminator molecule, ALDH1A3, which is postulated as a biomarker to select pro-invasive patients.

**Impact**

Our study proposes a new tool that could be used in the clinics to determine which patients would respond with increased cancer aggressiveness to antiangiogenic therapies.

test (2-tailed). Statistical comparison between control and treated mice was done by Mann–Whitney *t* test (2-tailed) or Chi-square test. Differences were considered statistically significant at $P < 0.05$. Actual $P$ and N values of each experiment are detailed in Appendix Table S5.

## Data availability

RNA sequencing generated in this study is readily available through Gene Expression Omnibus (GEO) GSE157802 and can be downloaded at https://www.ncbi.nlm.nih.gov/geo/query/acc.cgi?acc=GSE157802.

## Acknowledgements

The authors would like to thank Alba Martínez-López for expert technical support and Stefano Indraccolo (Veneto Institute of Oncology) for critical reading of the manuscript and helpful suggestions. pVHL-deficient 786-O (786-O) cell line of clear cell renal cell carcinoma was kindly provided by B. Jiménez (Instituto de Investigaciones Biomédicas CSIC-UAM, Madrid, Spain). This work was supported by research grants from ERC (ERC-StG-281830) EU-FP7, Ministerio de Ciencia, Innovación y Universidades, through the Retos Investigación grant, Refs: SAF2016-79347-R and PID2019-107557RB-I00/AEI/10.13039/501100011033, ISCIII Spain (AES, DTS17/00194), AGAUR-Generalitat de Catalunya (2017SGR771), Marató de TV3 (201910-30-31-32) and Italian Association for Cancer Research (AIRC) IG#14032, DOR - University of Padova, IOV 5x1000 fund. We thank CERCA Programme / Generalitat de Catalunya for institutional support.

## Author contributions

LM, RP, and GJ-V conduced in vivo experiments; GJ-V managed DC101 production and drug preparation; MM-L generated and maintained the orthoxenograft colony, supervised the in vivo experiments, and collected patient database; AV, CS, ET, JJ, IdT, and JC provided human specimens and clinical material; LM and RP realized immunohistochemistry and performed analysis on stained sections; LM, SA, JS, and LP managed molecular data from DNA and RNA sequencing; LM contributed to project conception and wrote the manuscript with input from other authors; AA critically commented the study and reviewed the manuscript; OC conceived the study, directed the project and supervised the writing and edited the manuscript.

## Conflict of interest

The authors declare that they have no conflict of interest.

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
