## [Review Process File · EMBO Molecular Medicine]

Kidney Cancer PDOXs reveal patient-specific pro-malignant effects of antiangiogenics and its molecular traits

Lidia Moserle, Roser Pons, Mar Martínez-Lozano, Gabriela Jiménez-Valerio, August Vidal, Cristina Suárez, Enrique Trilla, José Jiménez, Inés de Torres, Joan Carles, Jordi Senserrich, Susana Aguilar, Luis Palomero, Alberto Amadori, and Oriol Casanovas

DOI: [10.15252/emmm.201911889](https://doi.org/10.15252/emmm.201911889)

Corresponding author: Oriol Casanovas (ocasanovas@iconcologia.net)

Review Timeline:

Submission Date:	12th Dec 19
Editorial Decision:	29th Jan 20
Revision Received:	31st Jul 20
Editorial Decision:	1st Sep 20
Revision Received:	22nd Sep 20
Accepted:	25th Sep 20

Editor: Lise Roth

Transaction Report:

Thank you for the submission of your manuscript to EMBO Molecular Medicine, and please accept my apologies for the delay in getting back to you, which is due to the fact that referees needed more time to complete their review due to the holiday season. We have now received feedback from the two reviewers who agreed to evaluate your manuscript. As you will see from the reports below, the referees acknowledge the interest of the study and are overall supporting publication of your work pending major revisions.

Addressing the reviewers' concerns in full appears to require a lot of work, and a cross-commenting exercise helped us prioritize the points that should be addressed:

- 1/ concerns regarding the nature of the drug (TKI vs. bevacizumab),
- 2/ validation of the two-gene signature.

Adding further mechanistic insight on the other hand will not be necessary for further consideration in our journal. We realize that addressing the reviewers' concerns represents a lot of additional work, but given the interest of the findings, we are willing to extend our revision timeframe to six months instead of three if needed. Please be aware that acceptance of your revised manuscript will entail a second round of review.

EMBO Molecular Medicine encourages a single round of revision only and therefore, acceptance or rejection of the manuscript will depend on the completeness of your responses included in the next, final version of the manuscript. For this reason, and to save you from any frustrations in the end, I would strongly advise against returning an incomplete revision.

When submitting your revised manuscript, please carefully review the instructions that follow below. Failure to include requested items will delay the evaluation of your revision:

- 1) A .docx formatted version of the manuscript text (including legends for main figures, EV figures and tables). Please make sure that the changes are highlighted to be clearly visible.
- 2) Individual production quality figure files as .eps, .tif, .jpg (one file per figure).
- 3) A .docx formatted letter INCLUDING the reviewers' reports and your detailed point-by-point responses to their comments. As part of the EMBO Press transparent editorial process, the point-by-point response is part of the Review Process File (RPF), which will be published alongside your paper.
- 4) A complete author checklist, which you can download from our author guidelines (<https://www.embopress.org/page/journal/17574684/authorguide#submissionofrevisions>). Please insert information in the checklist that is also reflected in the manuscript. The completed author checklist will also be part of the RPF.
- 5) Before submitting your revision, primary datasets produced in this study need to be deposited in an appropriate public database (see

<https://www.embopress.org/page/journal/17574684/authorguide#dataavailability>).

The accession numbers and database should be listed in a formal "Data Availability" section (placed after Materials & Method). Please note that the Data Availability Section is restricted to new primary data that are part of this study.

6) We would also encourage you to include the source data for figure panels that show essential data. Numerical data should be provided as individual .xls or .csv files (including a tab describing the data). For blots or microscopy, uncropped images should be submitted (using a zip archive if multiple images need to be supplied for one panel). Additional information on source data and instruction on how to label the files are available at .

7) Our journal encourages inclusion of *data citations in the reference list* to directly cite datasets that were re-used and obtained from public databases. Data citations in the article text are distinct from normal bibliographical citations and should directly link to the database records from which the data can be accessed. In the main text, data citations are formatted as follows: "Data ref: Smith et al, 2001" or "Data ref: NCBI Sequence Read Archive PRJNA342805, 2017". In the Reference list, data citations must be labeled with "[DATASET]". A data reference must provide the database name, accession number/identifiers and a resolvable link to the landing page from which the data can be accessed at the end of the reference. Further instructions are available at .

8) We replaced Supplementary Information with Expanded View (EV) Figures and Tables that are collapsible/expandable online. A maximum of 5 EV Figures can be typeset. EV Figures should be cited as 'Figure EV1, Figure EV2' etc... in the text and their respective legends should be included in the main text after the legends of regular figures.

- Additional Tables/Datasets should be labeled and referred to as Table EV1, Dataset EV1, etc. Legends have to be provided in a separate tab in case of .xls files. Alternatively, the legend can be supplied as a separate text file (README) and zipped together with the Table/Dataset file. See detailed instructions here: .

9) The paper explained: EMBO Molecular Medicine articles are accompanied by a summary of the articles to emphasize the major findings in the paper and their medical implications for the non-specialist reader. Please provide a draft summary of your article highlighting

10) For more information: There is space at the end of each article to list relevant web links for further consultation by our readers. Could you identify some relevant ones and provide such information as well? Some examples are patient associations, relevant databases, OMIM/proteins/genes links, author's websites, etc...

11) Every published paper now includes a 'Synopsis' to further enhance discoverability. Synopses are displayed on the journal webpage and are freely accessible to all readers. They include a short stand first (maximum of 300 characters, including space) as well as 2-5 one-sentences bullet points that summarizes the paper. Please write the bullet points to summarize the key NEW findings. They should be designed to be complementary to the abstract - i.e. not repeat the same text. We encourage inclusion of key acronyms and quantitative information (maximum of 30 words / bullet point). Please use the passive voice. Please attach these in a separate file or send them by email, we will incorporate them accordingly.

Please also suggest a striking image or visual abstract to illustrate your article. If you do please provide a jpeg file 550 px-wide x 400-px high.

12) As part of the EMBO Publications transparent editorial process initiative (see our Editorial at <http://embomolmed.embopress.org/content/2/9/329>), EMBO Molecular Medicine will publish online a Review Process File (RPF) to accompany accepted manuscripts.

In the event of acceptance, this file will be published in conjunction with your paper and will include the anonymous referee reports, your point-by-point response and all pertinent correspondence relating to the manuscript. Let us know whether you agree with the publication of the RPF and as here, if you want to remove or not any figures from it prior to publication.

EMBO Molecular Medicine has a "scooping protection" policy, whereby similar findings that are published by others during review or revision are not a criterion for rejection. Should you decide to submit a revised version, I do ask that you get in touch after six months if you have not completed it, to update us on the status.

I look forward to receiving your revised manuscript.

Yours sincerely,

Lise Roth

Lise Roth, PhD
Editor
EMBO Molecular Medicine

To submit your manuscript, please follow this link:

Link Not Available

Photos 400-800 DPI

*Additional important information regarding figures and illustrations can be found at <http://bit.ly/EMBOPressFigurePreparationGuideline>

***** Reviewer's comments *****

Referee #1 (Remarks for Author):

Background

The contents of this manuscript deal primarily with a decade long controversial issue in the field of antiangiogenic therapy of cancer. In 2009 two preclinical papers were published in an issue of *Cancer Cell* in which both indicated that there are circumstances where an antiangiogenic drug may actually promote tumor progression, both local invasion and also metastatic disease. One of these papers is by the corresponding author of this present manuscript submitted to *EMM*, namely, Dr. Oriol Casanovas. Together the two *Cancer Cell* papers have been cited many thousands of times. There has been a number of preclinical studies undertaken since 2009 by other groups that have mostly confirmed that an antiangiogenic therapy can sometimes promote tumor progression, usually after initially showing an anti-tumor benefit. However, the major source of the controversy is related to the clinical relevance of these preclinical findings. In short, there is very little in the world's literature that indicates a worsened outcome such as exacerbation of metastatic disease or shortened survival when patients are treated with an antiangiogenic drug. This includes renal cell carcinoma (Blagoev et al "Sunitinib does not accelerate tumor growth in patients with metastatic renal cell carcinoma" *Cell Rep* 3:277-281, 2013). In any case, the authors of this study have probed this issue in an in-depth way by evaluating the impact of two different antiangiogenic drugs, one being DC101, an antibody directed against mouse VEGFR-2, and bevacizumab, the antibody to human VEGF. They did so by analyzing not only a particular RCC cell line (called 786-0) but much more importantly, by assessing the impact of these drugs on 56 different patient derived xenografts, or more accurately, on "PDOXs" (patient-derived orthotopic xenografts).

The major findings reported are that indeed, antiangiogenic drugs, or at least the two used, can induce/promote metastatic tumor progression in some of the PDOXs, but not in others. Interestingly, the pro-malignancy effect was preferentially observed in PDOXs that initially were minimally or non-invasive when the tumor tissue was orthotopically implanted into the kidney

capsule. In contrast, PDOXs that showed spontaneous capsular invasion at the outset, i.e., prior to therapy, tended not to show the increase in malignant aggressiveness when mice were treated with the antiangiogenic agents. Basically there was no change. Thus there seems to be intra-patient heterogeneity and variability with respect to the phenomenon, and that is one of the major take-home messages of the paper. The authors then undertook a genetic analysis to determine whether they could detect some changes that correlate with whether a particular PDOX responds with increased aggressiveness, or not, when the tumor bearing mice are treated with either antiangiogenic drug. They report changes in downstream genes related to K-ras signaling and also found a particular two gene signature, namely, ALDH1A3 (up) and MAP7 (down) by RNA sequencing. Of note, despite the increased aggressiveness that can be induced by the therapy, in general the mice show prolonged survival when treated with the antiangiogenic drugs, presumably because of suppression of primary tumor growth, which likely outweighs the increase in metastasis, many of which are micrometastases, located primarily in the lungs. Much of this analysis was done using four different PDOXs, two which became more metastatically aggressive, and two which did not, after antiangiogenic therapy.

Critique

This is a very interesting study with a number of notable features and novel findings. Clearly, one of the most obvious features is the tour de force effort in evaluating a large number of RCC PDOXs rather than one or two established RCC cell lines. To my knowledge, this kind of in-depth analysis, at least with respect to this phenomenon of drug-induced metastatic disease (by any drug), has not been undertaken previously. The finding that it is the minimally or non-invasive RCC PDOXs that show the increase in drug-induced malignant aggressiveness is interesting as are the gene expression changes that seem to correlate, and possibly predict the differential response of the PDOXs to antiangiogenic therapy. The use of the 'paired' PDOX model is also a strength where one member is metastatic and the partner is not. Altogether, this is an impressive body of preclinical work. Where I think the main problem arises is the clinical relevance of the findings. Let me explain.

First of all, as the authors are surely aware, there are numerous antiangiogenic drugs that are approved to treat RCC patients. Virtually all of these drugs are oral small molecule agents such as sunitinib, pazopanib, axitinib, etc. Antibodies to VEGFR-2 (e.g. ramucirumab) are not, to my knowledge, used to treat RCC patients. Bevacizumab is, but only in combination with interferon- α but is not, to my knowledge, commonly used. So an inevitable question is why the authors did not use a TKI such as sunitinib for their therapeutic studies rather than the two antibodies? This obvious translational point brings up another related one. If I understand the Tables correctly, the PDOXs were derived from patients who were therapy naïve. Presumably many of these patients were subsequently treated with standard-of-care antiangiogenic therapy (likely sunitinib) and if so, it is obvious to ask whether there was some sort of patient outcome correlation observed with the PDOX therapy results. For example, is it the case that the patients who 'donated' tumor tissue that showed increased aggressiveness in mice when the mice were treated with an antiangiogenic drug also show evidence of lack of response, i.e., disease progression as best response, or even increased metastatic disease when they were treated?

Returning to a point that was raised earlier, I believe it is the case that obvious increases in metastatic aggressiveness have not been reported in RCC patients receiving an antiangiogenic (TKI) therapy. About 80% of RCC patients show a clinical benefit, either partial responses, complete responses, or stable disease while the other 20% or so simply show disease progression as best response when treated with an antiangiogenic TKI such as sunitinib as first line therapy. So how do the authors explain this apparent discrepancy? Could it be that their findings are relevant to an

often observed finding that RCC patients derive a PFS benefit, but not an overall survival benefit? In other words, the possibility that there is an initial clinical benefit in PFS, which is subsequently eroded, at least in part, by a change in biologic aggressiveness so that no overall survival (OS) is observed? In this regard, it is important to note that the survival curves showed a benefit in the antiangiogenic drug-treated mice, even if such mice developed more metastases.

Technical issues/concerns

1. What strains of mice are used for this Ren-PDOX model, it's not mentioned at all in the Material and Method section. According to a paper (2016, Cell Reports 15, 1134-1143) published by the same group, nude mice were used for this model. How are the growth profiles of PDOXs? How long does it take for the tumor grafts to reach palpable sizes (1000mm³ according to the authors)? Two published papers reported very slow Ren-PDOX growth in NOD-SCIDs mice, one study from Sci Transl Med. 2012 June 6; 4(137) showed 1-8 months were required for the grafts to reach ~10mm in diameter (~500mm³), another study from J Pathol. 2011; 225(2): 212-221 showed 2-12months were needed to reach 2.4 to 29,452.5 mm³. In theory, Ren-PDOX growth will be much slower in nude mice than that in NOD-SCIDs mice; please provide detailed information.
2. How did the authors measure tumor volume? It is difficult to obtain accurate measurements of the orthotopically growing (internal) kidney tumor, posing a challenge in obtaining accurate tumor growth curves shown in Fig3. Please refer to a paper Sci Transl Med. 2012 June 6; 4(137) published by a team at University of Texas Southwestern Medical Center, who, using the Ren-PDOX model, found it was difficult to monitor the volumes of orthotopically growing tumor grafts (frequent imaging was needed to follow accurate tumor sizes), such that they switched to a more convenient subcutaneous model to do their therapy experiments.
3. What is the formula used in calculation of tumor volumes? According to Fig3, treatment was started when tumors were 1000mm³. What is the tumor's diameter? Was the tumor not too large for starting the treatment?
4. Fig2D showed three out of five Ren-PDOXs had point alterations in VHL, please provide specific information for Ren13, Ren28, Ren38, Ren50, Ren86-PDOX. Clinically, VHL-Hif axis is usually closely related to kidney cancer responsiveness to anti-angiogenic therapy.
5. How is the endpoint for the survival study in Fig3E defined? Is it based on sickness or tumor sizes? Ren50 reached endpoint at about day20 with tumor sizes at 1500mm³, while Ren28 tumor reached over 2500mm³ but was still the half way of its endpoint (see both Fig3A and Fig3E). Please explain the reason for this huge difference among the four Ren-PDOXs.
6. In Fig5, authors could apply the two-gene signature (ALDH1A3 up and MAP7 down) to Ren13BM (pro-invasive) and Ren28 (non-pro-invasive), which may provide direct evidence to show this two-gene signature has the power to predict pro-invasiveness of anti-angiogenic therapy.
7. Line300 on page8, "Briefly, whole-genome RNAseq of a pro-invasive PDOX model after anti-angiogenics (Ren13) and a non-pro-invasive PDOX (Ren50) in basal untreated condition was performed..." This description about sample selection is a source of confusion; isn't it a better comparison to use both Ren13 and Ren50 in basal untreated condition? Line314 on page9 stating this RNAseq was done on samples after anti-angiogenic treatment - this was also a source of confusion, as mentioned above. It makes more sense to compare these paired samples at basal untreated condition.

8. From Fig7 the message seems to be: the two-gene signature (ALDH1A3 up and MAP7 down) was indeed applied to define the tumor predisposition to acquire aggressive/invasive capacity in RCC patients, but not to define the responsiveness to anti-angiogenic therapy.

Additional questions that should be addressed or considered

1. Did authors test changes in invasive potential in PDOXs from non-metastatic patients rather than those with pre-existing increased invasiveness?
2. Fig 1G: need to clarify whether extent of invasion is representative of short term or survival experiment.
3. Fig 1H: assuming the data represents a long term experiment, could the increased extent of metastases be attributed to simply longer survival thus allowing more time for metastatic disease to manifest? Same question could be asked for Fig 1G.
4. Fig. 3: did REN28 and 50 show differential extent of metastatic potential (number of metastases) relative to REN13 and REN86? If so, could the difference in OS be attributed to lower invasive potential of the PDOX?
5. Fig 3E: need to clarify what was the primary cause of mortality -primary tumor invasion or metastatic disease?
6. Fig 5: how did REN86 and REN50 behave in terms of alterations/or lack thereof to metastatic potential under DC101 or BEV therapy? This information would be important to include.
7. Fig 5: low value N should be improved as it would help to strengthen the claims regarding alterations in invasive potential associated with DC and BEV.
8. Fig. 6: why was analysis done on REN50 and not REN28? The use of REN28 would keep consistency with data in Fig 5.
9. Fig 6: was analysis in Fig 6B conducted in DC101 or BEV treated tumors? It is unclear from the figure and legend. If no analysis was conducted with treated tumors, then it would be important to incorporate this information.

One other point of attribution: the term of "PDOX" was coined by Dr. Robert Hoffman and this should be cited. He has never received sufficient credit for his extensive body of PDOX studies despite publishing hundreds of papers on the technology (see Hoffman RM "Patient-derived orthotopic xenografts: better mimic of metastasis than subcutaneous xenografts" Nature Reviews Cancer 15: 451-452, 2015).

Conclusions and recommendation

Despite the breadth and nature of the concerns listed here, I am not inclined to recommend rejection, but rather to give the authors a chance to respond and consider submitting a revised manuscript. However, the revisions are major. While most of the technical concerns/questions can probably be easily addressed, the issue raised about the nature of the drugs used will require new experiments with an oral antiangiogenic TKI. Moreover, if the bevacizumab therapy experiments are

retained, I think an additional experiment involving the impact of bev plus interferon alpha would be necessary. There is another rationale for this for which there is a precedent. The rationale is that the increase in local invasion induced by bev monotherapy might be prevented by the interferon therapy. The precedent for this is a study by Paez-Ribes et al ("Potential pro-invasive or metastatic effects of preclinical antiangiogenic therapy are prevented by concurrent chemotherapy. Clin Cancer Res 21: 5488-5498, 2015) who showed that DC101 could increase local invasion of multiple orthotopic breast cancer xenografts - but this was prevented by concurrent chemotherapy using paclitaxel or cyclophosphamide.

Referee #2 (Comments on Novelty/Model System for Author):

The work is well done and most of it has significant breadth. However, parts of the study are limited in impact (See comments). The novelty is limited by prior publications from this group showing the effect of anti-angiogenic therapy in RCC PDXs. The potential impact of the work is high given the potential utility of a gene signature that could be exploited in the near-term. The models systems used are adequate.

Referee #2 (Remarks for Author):

Summary

Moserle et al investigated the heterogenous effects of antiangiogenic therapies on tumor invasiveness and metastasis with patient biopsy-derived orthotopic xenograft models and identified specific molecular biomarkers for selection of patients that might benefit from antiangiogenic therapies. They established an orthoxenograft RCC mouse model by orthotopically implanting 3D tumor pieces derived from intrakidney injection of RCC cells and defined invasive front as a measurement of local invasion. The authors further demonstrated that antiangiogenic strategies impaired tumor growth and extended overall survival but exacerbated tumor local invasion and metastatic dissemination in 786-O-derived xenografts. To investigate the effects of antiangiogenic therapy in a more clinically relevant model, they generated Ren-PDOXs by direct implantation of human tumor specimens from patients into mouse kidneys and confirmed these models recapitulated histological features and metastatic potential of the original patient tumor. Furthermore, they found Ren-PDOXs responded differently to antiangiogenic treatments regarding overall survival, which recapitulates the differential responses observed in patients. In the models that did not show survival benefits, antiangiogenic treatments increased the invasive capacity and systemic metastasis of these tumors. The authors performed RNAseq after anti-VEGF therapy from a pro-invasive PDOX model and a non-pro-invasive PDOX model and stratified tumor and stromal expression profiles. They found that pro-invasive tumors showed increased KRAS signaling compared to non-pro-invasive ones. They further filtered genes differentially expressed in RCC patients of more advanced/aggressive subsets and identified high ALDH1A3 and low MAP7 expression could be used to define the tumor predisposition to acquire aggressive potential in RCC patients.

Overall this is an interesting question to study patient-specific pro-malignant effects of antiangiogenic therapy and the authors provided supportive data. However, there are a few challenges that limit impact and should be addressed before further consideration.

General comments

1. PMID 2434631 & 27733559 should be added to the introduction section on fibrosis as they fit with and highlight that anti-VEGF therapy can increase fibrosis.

2. The Ren-PDOXs fully characterized here have been reported before (Jimenez-Valerio G, Cell Rep, 2016). The previous study also evaluated response and acquired resistance after VEGFR2 signaling inhibition with sunitinib, which limits the novelty of this study.

3. An overall concern is that the mechanism(s) is not clarified. The authors demonstrate the association of ALDH1A3 and MAP7 expression with invasive/aggressive tumor status but do not identify if/how these two genes contribute to response of antiangiogenic therapies. This is a deficiency especially given comment #2 above.

4. Having a predictive gene signature is potentially a significant step forward. Validation through TCGA analysis and gene expression correlated with vascular invasion in an independent patient cohort is useful and suggestive. However, in vivo validation using PDXs would greatly strengthen the conclusion.

Specific comments on figures

5. In Figure 1G and 1H, it is not clear whether those tissues were from short treatment or survival experiment.

6. In Figure 5, have the authors evaluated the level of circulating tumor cells post anti-VEGF therapy?

7. In Figure 6, the gene signature developed is limited due to the number of models in each category (n=1/category). While the differences are an exciting starting point, validation with additional models seems prudent (similar to comment 4 above)

Minor/editorial Comments:

Line 41: remove 'the'

Line 46: change 'to' to 'with'

Line 50: change 'metastatization' to 'metastasis'

Line 51: change 'tumor' to 'tumors'

Line 52: delete 'really'

Line 110: insert 'instead' after failed

Line 110: delete 'both'

Line 116: replace 'malignization' with 'progression'

Line 163: change 'metastatization' to 'metastasis'

Line 253: what does 'IF' mean here

Line 475: change 'metastatization' to 'metastasis'

8. The detection of only human VEGF-A in PDOXs is not consistent with the literature that stromal cells produce VEGF-A in the tumor microenvironment. Please discuss.

9. Therapy was initiated when the tumors were large - 1000 mm³ is a considerable size for a mouse. What happens if you start therapy earlier, if the tumor is 200 mm³ when it is palpable and angiogenic but likely not necrotic?

Moserle et al. EMM-2019-11889
RESPONSE TO REFEREE'S CRITIQUES

Referee #1 (Remarks for Author):

Background

The contents of this manuscript deal primarily with a decade long controversial issue in the field of antiangiogenic therapy of cancer. In 2009 two preclinical papers were published in an issue of Cancer Cell in which both indicated that there are circumstances where an antiangiogenic drug may actually promote tumor progression, both local invasion and also metastatic disease. One of these papers is by the corresponding author of this present manuscript submitted to EMM, namely, Dr. Oriol Casanovas. Together the two Cancer Cell papers have been cited many thousands of times. There has been a number of preclinical studies undertaken since 2009 by other groups that have mostly confirmed that an antiangiogenic therapy can sometimes promote tumor progression, usually after initially showing an anti-tumor benefit. However, the major source of the controversy is related to the clinical relevance of these preclinical findings. In short, there is very little in the world's literature that indicates a worsened outcome such as exacerbation of metastatic disease or shortened survival when patients are treated with an antiangiogenic drug. This includes renal cell carcinoma (Blagoev et al "Sunitinib does not accelerate tumor growth in patients with metastatic renal cell carcinoma" Cell Rep 3:277-281, 2013). In any case, the authors of this study have probed this issue in an in-depth way by evaluating the impact of two different antiangiogenic drugs, one being DC101, an antibody directed against mouse VEGFR-2, and bevacizumab, the antibody to human VEGF. They did so by analyzing not only a particular RCC cell line (called 786-0) but much more importantly, by assessing the impact of these drugs on 56 different patient derived xenografts, or more accurately, on "PDOXs" (patient-derived orthotopic xenografts).

The major findings reported are that indeed, antiangiogenic drugs, or at least the two used, can induce/promote metastatic tumor progression in some of the PDOXs, but not in others. Interestingly, the pro-malignancy effect was preferentially observed in PDOXs that initially were minimally or non-invasive when the tumor tissue was orthotopically implanted into the kidney capsule. In contrast, PDOXs that showed spontaneous capsular invasion at the outset, i.e., prior to therapy, tended not to show the increase in malignant aggressiveness when mice were treated with the antiangiogenic agents. Basically there was no change. Thus there seems to be intra-patient heterogeneity and variability with respect to the phenomenon, and that is one of the major take-home messages of the paper. The authors then undertook a genetic analysis to determine whether they could detect some changes that correlate with whether a particular PDOX responds with increased aggressiveness, or not, when the tumor bearing mice are treated with either antiangiogenic drug. They report changes in downstream genes related to K-ras signaling and also found a particular two gene signature, namely, ALDHIA3 (up) and MAP7 (down) by RNA sequencing. Of note, despite the increased aggressiveness that can be induced by the therapy, in general the mice show prolonged survival when treated with the antiangiogenic drugs, presumably because of suppression of primary tumor growth, which likely outweighs the increase in metastasis, many of which are micrometastases, located

primarily in the lungs. Much of this analysis was done using four different PDOXs, two which became more metastatically aggressive, and two which did not, after antiangiogenic therapy.

Critique

This is a very interesting study with a number of notable features and novel findings. Clearly, one of the most obvious features is the tour de force effort in evaluating a large number of RCC PDOXs rather than one or two established RCC cell lines. To my knowledge, this kind of in-depth analysis, at least with respect to this phenomenon of drug-induced metastatic disease (by any drug), has not been undertaken previously. The finding that it is the minimally or non-invasive RCC PDOXs that show the increase in drug-induced malignant aggressiveness is interesting as are the gene expression changes that seem to correlate, and possibly predict the differential response of the PDOXs to antiangiogenic therapy. The use of the 'paired' PDOX model is also a strength where one member is metastatic and the partner is not. Altogether, this is an impressive body of preclinical work. Where I think the main problem arises is the clinical relevance of the findings. Let me explain.

First of all, as the authors are surely aware, there are numerous antiangiogenic drugs that are approved to treat RCC patients. Virtually all of these drugs are oral small molecule agents such as sunitinib, pazopanib, axitinib, etc. Antibodies to VEGFR-2 (e.g. ramucirumab) are not, to my knowledge, used to treat RCC patients. Bevacizumab is, but only in combination with interferon- α but is not, to my knowledge, commonly used. So an inevitable question is why the authors did not use a TKI such as sunitinib for their therapeutic studies rather than the two antibodies?

We thank very much this referee for pointing out the novelty of our study and putting it into perspective with the background in the field.

To directly answer this first question, initially the study was designed to use the most specific antiangiogenics that blocked the ligand or the receptor of VEGF, not to have "off-target" effects of the current TKIs used in the clinic. Indeed, we agree with the referee that clinically it makes a lot more sense to use TKIs such as sunitinib. Thus, we have performed 7 new in vivo experiments using sunitinib as clinically-meaningful antiangiogenic, which we include in this revised manuscript. **New Figure 4D** describes capsular and local invasion effects of sunitinib treatment in a pro-invasive and a non-pro-invasive PDOX, which are consistent with previous results observed with DC101 or Bevacizumab in these same PDOXs. Furthermore, sunitinib treatment study has been expanded to 5 new Ren-PDOXs, where pro-invasive response has been observed in 3 of them, results included in **new Figure EV3D**.

This obvious translational point brings up another related one. If I understand the Tables correctly, the PDOXs were derived from patients who were therapy naïve. Presumably many of these patients were subsequently treated with standard-of-care antiangiogenic therapy (likely sunitinib) and if so, it is obvious to ask whether there was some sort of patient outcome correlation observed with the PDOX therapy results. For example, is it the case that the patients who 'donated' tumor tissue that showed increased aggressiveness in mice when the mice were treated with an antiangiogenic drug also show evidence of lack of response, i.e., disease progression as best response, or even increased metastatic disease when they were

treated?

This is a very important question and we have tried to address this. While the number of PDOXs that we have is too low to produce statistics, we have observed some consistency between PDOX pro-malignancy and patient metastasis formation after treatment. For example, Ren13 responded pro-aggressively in PDOX and the patient progressed with brain metastasis after antiangiogenic therapy was administered in the clinics. On the other hand, in the case of Ren50, the patient was already metastatic when biopsy was obtained, so we cannot conclude whether treatment induced more aggressiveness or not. As for Ren 28 and Ren 86, aggressive response to treatment cannot be determined because one patient died before treatment, and the other started antiangiogenics, but had to discontinue due to toxicity.

While interesting, we have decided not to include this information to the manuscript, unfortunately.

Returning to a point that was raised earlier, I believe it is the case that obvious increases in metastatic aggressiveness have not been reported in RCC patients receiving an antiangiogenic (TKI) therapy. About 80% of RCC patients show a clinical benefit, either partial responses, complete responses, or stable disease while the other 20% or so simply show disease progression as best response when treated with an antiangiogenic TKI such as sunitinib as first line therapy. So how do the authors explain this apparent discrepancy? Could it be that their findings are relevant to an often observed finding that RCC patients derive a PFS benefit, but not an overall survival benefit? In other words, the possibility that there is an initial clinical benefit in PFS, which is subsequently eroded, at least in part, by a change in biologic aggressiveness so that no overall survival (OS) is observed? In this regard, it is important to note that the survival curves showed a benefit in the antiangiogenic drug-treated mice, even if such mice developed more metastases.

Again, a very clinically relevant point is raised by this reviewer, which we thank so much.

Our data indicates that this apparent discrepancy is indeed related to a direct antitumor benefit in all PDXs tested, but a differential benefit in OS in proinvasive or non-proinvasive tumors. Specifically, as shown in **Figure 3E**, OS benefit is achieved in Ren28 and Ren50 (non-proinvasive PDOXs), while in pro-invasive PDOXs (Ren13 and Ren86) there is no benefit in OS. Therefore, our data are consistent with the hypothesis that antiangiogenics produce a PFS benefit in a majority of patients, but the pro-invasive response in some of them erodes this benefit into no OS benefit.

This is an important point, so we have included it in the **discussion (page 12)**.

Technical issues/concerns

1. What strains of mice are used for this Ren-PDOX model, it's not mentioned at all in the Material and Method section. According to a paper (2016, Cell Reports 15, 1134-1143) published by the same group, nude mice were used for this model. How are the growth profiles of PDOXs? How long does it take for the tumor grafts to reach palpable sizes (1000mm³ according to the authors)? Two published papers reported very slow Ren-PDOX growth in NOD-SCIDs mice, one study from Sci Transl Med. 2012 June 6; 4(137) showed 1-8 months were required for the grafts to reach ~10mm in diameter (~500mm³), another study

from J Pathol. 2011; 225(2): 212-221 showed 2-12months were needed to reach 2.4 to 29,452.5 mm³. In theory, Ren-PDOX growth will be much slower in nude mice than that in NOD-SCIDs mice; please provide detailed information.

Ren-PDOX models were developed in nude mice. Our method of implantation is to stitch a 3D piece of intact tissue (no mincing or grinding), so the tissue architecture is preserved upon implantation and subsequent growth of the PDOX. This is an important technical difference from many of the PDX and PDOX published in the literature where tissue disaggregation is typically performed. This is a very important difference, as we have tested in our hands that, with the exact same “donor” tumor and host, tissue grinding, disaggregation or digestion produces a much slower growth than intact tissue piece implantation. Indeed, compared to previous studies mentioned, our PDOX models take an average of two months to grow until we can start experiments (median 53 days CI95% [40-95days]).

For its important differences with other previously published procedures, we have included the details of our methodology in **Materials and Methods section (page 13)**.

2. How did the authors measure tumor volume? It is difficult to obtain accurate measurements of the orthotopically growing (internal) kidney tumor, posing a challenge in obtaining accurate tumor growth curves shown in Fig3. Please refer to a paper Sci Transl Med. 2012 June 6; 4(137) published by a team at University of Texas Southwestern Medical Center, who, using the Ren-PDOX model, found it was difficult to monitor the volumes of orthotopically growing tumor grafts (frequent imaging was needed to follow accurate tumor sizes), such that they switched to a more convenient subcutaneous model to do their therapy experiments.

Indeed the evaluation of tumor growth in peritoneal tumors is a challenge, but we have to honestly say that we made huge efforts and succeeded to develop a robust procedure to quantify this when we started our tour de force effort of PDOX implantation, back in 2011. To try to overcome this challenge, we initially made use of a luciferase-transfected 786O- tumor model which was passaged using the PDOX protocol (3D tumor piece implantation) and we developed a scale of tumor volume measurements by palpation with their corresponding luciferin voxel volumes and the actual volumes (by surgical excision of the tumor specimen at defined time points in different animals).

Furthermore, we made use of RMN to evaluate tumor volume in Ren-PDOXs to refine our scale of tumor volumes, and with the final tumor excision volume, we were able to generate a robust scale of tumor volumes defined in mm³.

RMN T2 images:

RMN water diffusion map:

We have to say that in a few infrequent cases, we found discrepancies between our tumor volume method and the actual final tumor weight, but these were particular cases where a liquid cyst had grown close to the tumor (hydronephrosis). This liquid accumulation produced a false increase of volume in our method, but was not really tumor mass growing in the kidney. These rare cases were obviously excluded from any experiment performed, of course.

Even though we are very confident that our tumor volume method is valid, in all our studies we Always validate our tumor volume data with the final tumor weight evaluation as a hard number measurement (i.e. Figure 3B where final tumor weight is shown).

To clarify this, we have included a description of tumor volume measurement in **Materials and Methods section (page 13)**, thank you.

3. What is the formula used in calculation of tumor volumes? According to Fig3, treatment was started when tumors were 1000mm³. What is the tumor's diameter? Was the tumor not too large for starting the treatment?

Tumor volumes shown in Figure 3A were calculated using our scale of tumor volume (see previous question), which were referenced to RMN 3D volume calculations (in mm³) and validated by excised tumor volume (calculated by sphere volume formula). Approximate starting diameter was 12mm. Indeed this is a bigger starting tumor volume compared to many other tumor studies (much bigger than subcutaneous models, and bigger than orthotopic cell-injection models). This is explained by the fact that we implant a piece of tissue that already has 3-5mm of diameter which is much bigger than many other protocols. Thus, starting at 3-5 mm diameter, plus the tissue damage and scarring that a stitch implantation produces, we felt confident in starting treatment when the tumor is 12mm (double of starting size), when we are sure it is already growing at exponential capacity, and more importantly it has developed its vasculature to nurture the growing tissue and there is no necrosis in the tumor mass (specifically checked by H&E staining and Tunel, data not shown).

4. Fig2D showed three out of five Ren-PDOXs had point alterations in VHL, please provide specific information for Ren13, Ren28, Ren38, Ren50, Ren86-PDOX. Clinically, VHL-Hif axis is usually closely related to kidney cancer responsiveness to anti-angiogenic therapy.

VHL mutational status for the five Ren-PDOXs has been included in **Appendix TableS4**. Specifically, Ren13, 28 & 38 show VHL mutation while Ren50 & Ren86 do not. Thus, we conclude that detected VHL mutations do not seem to be associated with pro-aggressive response to treatment as pro-invasive PDOXs (Ren13 and Ren86, one with detected mutation but the other not detected).

5. How is the endpoint for the survival study in Fig3E defined? Is it based on sickness or tumor sizes? Ren50 reached endpoint at about day20 with tumor sizes at 1500mm³, while Ren28 tumor reached over 2500mm³ but was still the half way of its endpoint (see both Fig3A and Fig3E). Please explain the reason for this huge difference among the four Ren-PDOXs.

In survival experiments, mice are sacrificed according to strict health welfare criteria, not tumor volume. Being a peritoneal tumor, volume is not as limiting as in subcutaneous models, and physiologically allows for more volume and organ displacement (i.e. pregnancy). This means that there are tumors that with a small volume already make the animals more symptomatic (and sick), while in other tumors the animals survive with a higher volume without symptoms. This is a characteristic of each tumor (and patient), and while it shows variability between different PDOXs, it is fairly stable among different animals of the same PDOX. In the case of Ren50, indeed OS is short with lower tumor volumes. We suspect that this tumor is highly secretory of cytokines or other peptides that make the host animal get sick quite quickly compared to other PDOXs.

We have included our end-point criteria in **Materials and Methods section, page 14**.

6. In Fig5, authors could apply the two-gene signature (ALDH1A3 up and MAP7 down) to Ren13BM (pro-invasive) and Ren28 (non-pro-invasive), which may provide direct evidence to show this two-gene signature has the power to predict pro-invasiveness of anti-angiogenic therapy.

We thank the referee for this important critique that coincides with Referee #2 and the Editor. We have done extensive work to expand this part of our study with several new response prediction experiments (**full new Figures 7 and EV5**). Initially, as suggested by this referee, we analyzed our PDOX samples by mRNA and IHC and found that ALDH1A3 can clearly discriminate which are the pro-invasive models, but MAP7 results were not consistent (**new Figure 7A&B and EV5A**). Puzzled by these data, we extended our IHC study to the new sunitinib-treated Ren-PDOXs to find that ALDH1A3 is a clear and statistical significant discriminator while MAP7 fails to even associate to invasion phenotype (**new Figure 7C&D and EV5B, C & D**).

Thus, we decided to go for a final prediction validation using a new series of RCC patients treated with antiangiogenics. We gathered a series of 15 patients clinically treated with sunitinib whose pro-aggressive response after therapy was fully annotated, including disease progression with new metastasis or overt local infiltration, vs disease progression with no new lesions or local progression (**new Figure 7E**). Tumor specimen analysis of this series not only confirmed that ALDH1A3 can significantly discriminate pro-aggressive response in patients, but also allowed us to perform a ROC curve study to define predictive power of ALDH1A3, which shows very significant results (**new Figure 7F, G & H**).

Overall, we thank this referee for having pushed us to extend our study which has clearly strengthened and improved our manuscript with this better-tested predictive factor of pro-aggressiveness.

7. Line300 on page8, "Briefly, whole-genome RNAseq of a pro-invasive PDOX model after anti-angiogenics (Ren13) and a non-pro-invasive PDOX (Ren50) in basal untreated condition was performed..." This description about sample selection is a source of confusion; isn't it a better comparison to use both Ren13 and Ren50 in basal untreated condition? Line314 on page9 stating this RNAseq was done on samples after anti-angiogenic treatment - this was also a source of confusion, as mentioned above. It makes more sense to compare these paired samples at basal untreated condition.

We thank this referee to allow us to clarify this confusion. Indeed it is a nomenclature confusion and the analyzed samples are all pre-treatment. The cause of this confusion is in wrongly stating the type of response as "pro-invasive after antiangiogenics", which got confused with samples treated with antiangiogenics. We have amended this by omitting the "after antiangiogenics" detail to clarify that tumor samples are pre-treatment. Thank you.

8. From Fig7 the message seems to be: the two-gene signature (ALDH1A3 up and MAP7 down) was indeed applied to define the tumor predisposition to acquire aggressive/invasive capacity in RCC patients, but not to define the responsiveness to anti-angiogenic therapy.

We agree with this referee that our initial validation was using non-treated patient samples, and this had to be improved. Thus, as explained in detail in question 6, we have expanded this part of the manuscript to truly test the predictive value of ALDH1A3 in response to anti-angiogenic therapy. To do that, we not only used 7 sunitinib treated Ren-PDOXs but also a new series of 15 patient samples with full annotation of their progression characteristics. As extensively explained in response to question #6, these new important data are now presented in **full new Figure 7** and **new Figure EV5**.

Additional questions that should be addressed or considered

1. Did authors test changes in invasive potential in PDOXs from non-metastatic patients rather than those with pre-existing increased invasiveness?

Yes indeed. As shown in Appendix Table S1, Ren-PDOXs were developed from donor RCC patients irrespective of their M-stage (metastasis or not). Specifically, 17 out of 26 (65%) engrafted PDOXs were non-metastatic and 9 out of 26 (35%) were metastatic.

We evaluated pro-invasive potential in both metastatic and non-metastatic patient-derived models. For example, Ren13 PDOX was derived from a donor patient that did not present metastasis at the time of donation and PDOX establishment.

2. Fig 1G: need to clarify whether extent of invasion is representative of short term or survival experiment.

We thank this reviewer for allowing us to clarify this **in the figure legend**. Fig 1G is from a short-term experiment, as this pro-invasive phenomenon is observed quite early upon treatment.

3. Fig 1H: assuming the data represents a long term experiment, could the increased extent of metastases be attributed to simply longer survival thus allowing more time for metastatic disease to manifest? Same question could be asked for Fig 1G.

It is indeed possible that differences in survival could increment the quantity of metastasis in our survival experimental design. Nevertheless, the main point of this study is that while part of the models increase their metastasis, others do not increase metastasis (even there is a survival extension) see Figure 3E. On the other hand, data in Figure 1G does not have any difference in survival, because it is a short-term experiment (as explained in the previous question).

We have clarified this **in the figure legend and text**, thank you.

4. Fig. 3: did REN28 and 50 show differential extent of metastatic potential (number of metastases) relative to REN13 and REN86? If so, could the difference in OS be attributed to lower invasive potential of the PDOX?

This is an intriguing hypothesis, but our data does not seem to back this up. In fact, our most prominent metastatic models are Ren13 and Ren28. Of these two, one is pro-invasive and does not show OS benefit (Ren13), but the other is not pro-invasive and does show OS benefit. Thus, differences in OS do not seem to be related to invasive/metastatic capacity of each Ren-PDOX.

5. Fig 3E: need to clarify what was the primary cause of mortality -primary tumor invasion or metastatic disease?

In our experiments, the primary cause of mortality does not seem to be metastatic disease but rather primary tumor progression. In fact, not all PDOX models are metastatic, and when metastasis occurs, it is typically limited to a few lesions that likely do not impair lung functionality (see H&E images in Figure 5A). So primary tumor progression both in tumor mass and tumor local infiltration seem to be the cause of mortality. Furthermore, RCC tumors are

known to have hypersecretory capacity and secrete cytokines or other peptides that make the host animal get sick quite quickly, thus limiting their survival. In fact, this seems to be the case of Ren50, where OS is short and animals have to be sacrificed when they have not so big tumor volumes, and we suspect that this tumor is highly secretory of compared to other PDOXs.

We have included this information in **Materials and Methods section, page 14**.

6. Fig 5: how did REN86 and REN50 behave in terms of alterations/or lack thereof to metastatic potential under DC101 or BEV therapy? This information would be important to include.

As mentioned in a previous question and in the Appendix Tables, not all PDOX models are metastatic. In fact, the only overtly metastatic models in our treatment study are the ones already included in Figure 5. Thus, we did not include Ren86 or Ren50 in this figure because these two models do not produce reproducible (and quantifiable) metastasis, unfortunately. In order to generate metastasis in these models, we even tried to do nephrectomies (tumor and kidney excision) to extend survival in these animals allow for more time to develop metastasis. Unfortunately, even with nephrectomies these models did not produce reproducible and quantifiable metastasis, so we did not include them in the metastasis study in Figure 5.

7. Fig 5: low value N should be improved as it would help to strengthen the claims regarding alterations in invasive potential associated with DC and BEV.

We agree with this reviewer about the low N. We have performed a new in vivo experiment and we have increased the number of animals included in this Figure. Thank you very much.

8. Fig. 6: why was analysis done on REN50 and not REN28? The use of REN28 would keep consistency with data in Fig 5.

The reviewer is right, Ren28 would have kept better consistency. Nevertheless, we did the molecular analysis on Ren50 because we wanted to minimize inter-patient variability (to produce less “noise” in expression comparison) and Ren13 and Ren50 have more similar characteristics in terms of clear cell cellularity and sarcomatoid phenotype. Sarcomatoid differentiation in RCC is a tumor cell histological characteristic associated to bad prognosis and functionally related to more aggressive disease and propensity for metastasis (Mouallem et al. 2018 doi: 10.1016/j.urolonc.2017.12.012, PMID: 29306556).

As Ren28 shows less clear cell phenotype and a 70% sarcomatoid transformation (as per anatomopathology diagnostic report), in the molecular analysis it could have generated more “false positives” related to cellular phenotype differences than their response to treatment. Thus, we decided to use Ren50 because it has “pure” clear cell phenotype without sarcomatoid differentiation, exactly as Ren13.

9. Fig 6: was analysis in Fig 6B conducted in DC101 or BEV treated tumors? It is unclear from the figure and legend. If no analysis was conducted with treated tumors, then it would be important to incorporate this information.

Aiming at defining a predictive factor of this pro-invasive effect, we explicitly chose to molecularly analyze pre-treatment samples. As clarified in **the text (page 9)**, samples before treatment were used for the expression analysis in search of basal pre-treatment differences

that could pre-define or predict the subsequent differential response to treatment. Purely a predictive factor of response.

One other point of attribution: the term of "PDOX" was coined by Dr. Robert Hoffman and this should be cited. He has never received sufficient credit for his extensive body of PDOX studies despite publishing hundreds of papers on the technology (see Hoffman RM "Patient-derived orthotopic xenografts: better mimic of metastasis than subcutaneous xenografts" Nature Reviews Cancer 15: 451-452, 2015).

Indeed this referee is right that we should give much more credit to the "father" of PDOXs. We have included this citation in the **Introduction section, page 4**. Thank you.

Conclusions and recommendation

Despite the breadth and nature of the concerns listed here, I am not inclined to recommend rejection, but rather to give the authors a chance to respond and consider submitting a revised manuscript. However, the revisions are major. While most of the technical concerns/questions can probably be easily addressed, the issue raised about the nature of the drugs used will require new experiments with an oral antiangiogenic TKI. Moreover, if the bevacizumab therapy experiments are retained, I think an additional experiment involving the impact of bev plus interferon alpha would be necessary. There is another rationale for this for which there is a precedent. The rationale is that the increase in local invasion induced by bev monotherapy might be prevented by the interferon therapy. The precedent for this is a study by Paez-Ribes et al ("Potential pro-invasive or metastatic effects of preclinical antiangiogenic therapy are prevented by concurrent chemotherapy. Clin Cancer Res 21: 5488-5498, 2015) who showed that DC101 could increase local invasion of multiple orthotopic breast cancer xenografts - but this was prevented by concurrent chemotherapy using paclitaxel or cyclophosphamide.

Thank you very much for detailed critiques and insightful comments (and final conclusion) which have helped us to greatly improve our manuscript. Thank you.

Referee #2 (Remarks for Author):

Summary

Moserle et al investigated the heterogeneous effects of antiangiogenic therapies on tumor invasiveness and metastasis with patient biopsy-derived orthotopic xenograft models and identified specific molecular biomarkers for selection of patients that might benefit from antiangiogenic therapies. They established an orthoxenograft RCC mouse model by orthotopically implanting 3D tumor pieces derived from intrakidney injection of RCC cells and defined invasive front as a measurement of local invasion. The authors further demonstrated that antiangiogenic strategies impaired tumor growth and extended overall survival but exacerbated tumor local invasion and metastatic dissemination in 786-O-derived xenografts. To investigate the effects of antiangiogenic therapy in a more clinically relevant model, they generated Ren-PDOXs by direct implantation of human tumor specimens from patients into mouse kidneys and confirmed these models recapitulated histological features and metastatic potential of the original patient tumor. Furthermore, they found Ren-PDOXs responded differently to antiangiogenic treatments regarding overall survival, which recapitulates the differential responses observed in patients. In the models that did not show survival benefits, antiangiogenic treatments increased the invasive capacity and systemic metastasis of these tumors. The authors performed RNAseq after anti-VEGF therapy from a pro-invasive PDOX model and a non-pro-invasive PDOX model and stratified tumor and stromal expression profiles. They found that pro-invasive tumors showed increased KRAS signaling compared to non-pro-invasive ones. They further filtered genes differentially expressed in RCC patients of more advanced/aggressive subsets and identified high ALDH1A3 and low MAP7 expression could be used to define the tumor predisposition to acquire aggressive potential in RCC patients.

Overall this is an interesting question to study patient-specific pro-malignant effects of antiangiogenic therapy and the authors provided supportive data. However, there are a few challenges that limit impact and should be addressed before further consideration.

General comments

1. PMID 2434631 & 27733559 should be added to the introduction section on fibrosis as they fit with and highlight that anti-VEGF therapy can increase fibrosis.

We have included these two citations in the **Introduction section, page 3**. Thank you.

2. The Ren-PDOXs fully characterized here have been reported before (Jimenez-Valerio G, Cell Rep, 2016). The previous study also evaluated response and acquired resistance after VEGFR2 signaling inhibition with sunitinib, which limits the novelty of this study.

This referee is right that 2 of our models were used in a previous study to define a specific mechanism of resistance to anti-angiogenics. Nevertheless the breadth of the current study in terms of number of models, characterization and the specific evaluations is much more extensive, and focused to answer a very different question. Indeed, the focus of this study is to evaluate pro-invasion and metastasis (which were not characterized in the previous study) and to molecularly pinpoint a biomarker to putatively discriminate these distinct responses.

We respectfully disagree with this referee as we are convinced this is a novel study.

3. An overall concern is that the mechanism(s) is not clarified. The authors demonstrate the association of ALDH1A3 and MAP7 expression with invasive/aggressive tumor status but do not identify if/how these two genes contribute to response of antiangiogenic therapies. This is a deficiency especially given comment #2 above.

We agree with this referee that mechanism is not clarified. Nevertheless, the aim and focus of this study was related to describing whether there was a differential response to antiangiogenics and how could we open avenues for detection or discrimination of these tumors against the rest. Pinpointing the mechanism would have been very interesting indeed; nevertheless we focused on finding a predictive biomarker of pro-invasive response as an applicable avenue of our study for the patients in the future.

We hope this referee will understand our interest and focus in prediction rather than cause.

4. Having a predictive gene signature is potentially a significant step forward. Validation through TCGA analysis and gene expression correlated with vascular invasion in an independent patient cohort is useful and suggestive. However, in vivo validation using PDXs would greatly strengthen the conclusion.

We thank the referee for this important critique that coincides with Referee #1 and the Editor. We fully agree with this comment and we have done extensive work to expand this part of our study with several new response prediction experiments (**full new Figures 7 and EV5**). Initially, as suggested by this referee, we analyzed our PDOX samples by mRNA and IHC and found that ALDH1A3 can clearly discriminate which are the pro-invasive models, but MAP7 results were not consistent (**new Figure 7A&B and EV5A**). Puzzled by these data, we extended our IHC study to the new sunitinib-treated Ren-PDOXs to find that ALDH1A3 is a clear and statistical significant discriminator while MAP7 fails to even associate to invasion phenotype (**new Figure 7C&D and EV5B, C &D**).

Thus, we decided to go for a final prediction validation using RCC patients' samples: We gathered a series of 15 patients clinically treated with sunitinib whose pro-aggressive response after therapy was fully annotated, including disease progression with new metastasis or overt local infiltration, vs disease progression with no new lesions or local progression (**new Figure 7E**). Tumor specimen analysis of this series not only confirmed that ALDH1A3 can significantly discriminate pro-aggressive response in patients, but also allowed us to perform an initial ROC curve study to define predictive power of ALDH1A3, which shows very significant results (**new Figure 7F, G & H**).

Overall, we thank this referee for having pushed us to extend our study and clearly strengthen and improved our manuscript with this better-tested predictive factor of pro-aggressiveness.

Specific comments on figures

5. In Figure 1G and 1H, it is not clear whether those tissues were from short treatment or survival experiment.

We thank this referee for this comment. To clarify, Fig 1G is from a short-term experiment, as this pro-invasive phenomenon is observed quite early upon treatment. On the other hand, Figure 1H is from a long term (survival) experiment, where the late event of metastasis can be observed. We have clarified this **in the figure legend and text (page 5)**. Thank you.

6. In Figure 5, have the authors evaluated the level of circulating tumor cells post anti-VEGF therapy?

We thank the referee for this thoughtful comment. Indeed, we tried to detect circulating tumor cells as it would have been a very important aspect to understand pro-metastatic effects of antiangiogenics. We tried the Veridex machine but the expression of EpCAM (fixed marker) in these RCC cells is extremely low, and other surface tumor markers were heterogeneous (Cytokeratin AE1/AE3...). We also tried by FlowCytometry, but the signal was not robust enough to produce meaningful data. At the end, unfortunately, we were unable to produce convincing evidence of true RCC circulating cells, so we did not include these experiments in the manuscript.

Thank you very much for this comment.

7. In Figure 6, the gene signature developed is limited due to the number of models in each category ($n=1/\text{category}$). While the differences are an exciting starting point, validation with additional models seems prudent (similar to comment 4 above)

We fully agree with this referee as this was an initial limitation to this analysis. Very aware of this constraint, we subsequently did a filtering through 528 patients' data from TCGA in order to eliminate specific differences between these two particular tumors and obtain more relevant data applicable to a broader range of RCC patients. Moreover, we validated this with the old invasion samples series, and we have now further validated its predictive value using more PDOX and a new series of 15 clinical samples (**whole new Figure 7 and Figure EV5**). Overall, while the initial analysis is restricted to two models, the several subsequent steps of filtering and validation produce at the end meaningful and validated results, which we think could be significant for RCC patients in general.

We truly thank this referee as this comment has made us greatly improve our manuscript.

Minor/editorial Comments:

Line 41: remove 'the'

Line 46: change 'to' to 'with'

Line 50: change 'metastatization' to 'metastasis'

Line 51: change 'tumor' to 'tumors'

Line 52: delete 'really'

Line 110: insert 'instead' after failed

Line 110: delete 'both'

Line 116: replace 'malignization' with 'progression'

Line 163: change 'metastatization' to 'metastasis'

Line 253: what does 'IF' mean here

Line 475: change 'metastatization' to 'metastasis'

All these mistakes were amended, and IF was replaced by "invasive front". Thank you.

8. The detection of only human VEGF-A in PDOXs is not consistent with the literature that stromal cells produce VEGF-A in the tumor microenvironment. Please discuss.

We fully agree with this comment because we know stromal VEGF-A is present in our PDOX models, but at extremely lower levels than tumor derived (human) VEGF-A. This is due to the

biology of RCC where VHL loss produces a genetically-driven accumulation of HIF1a-dependent genes, including VEGF-A. Furthermore, the proportion of stromal (mouse) tissue in a defined piece of tumor is much smaller than human tumor tissue (9-12% in our studies, data not shown). Thus, in the species-specific RNA-seq study (Figure EV2), mouse VEGF-A is present but at a minimal level when measured in “counts per million” (Cpm) and cannot be observed in the graph.

We clarified this in the **text (page 6)**.

9. Therapy was initiated when the tumors were large - 1000 mm³ is a considerable size for a mouse. What happens if you start therapy earlier, if the tumor is 200 mm³ when it is palpable and angiogenic but likely not necrotic?

All in vivo experiments started at large 1000 mm³, which correspond to approximate starting diameter of 12mm. Indeed this is a bigger starting tumor volume compared to many other tumor studies (much bigger than subcutaneous models, and bigger than orthotopic models). This is explained by the fact that we implant a piece of tissue that already has 3mm of diameter which is much bigger than many other protocols. Thus, starting at 3mm diameter, plus the tissue damage and scarring that a stitch implantation produces, we felt confident in starting treatment when the tumor is 12mm, when we are sure it is already growing at quasi-exponential capacity.

Regarding necrosis, in general, orthotopic implanted tumors grow from a better vascularized tissue and show much less necrosis at a higher tumor volume than subcutaneous tumors, as reported by Fleming et al. 2010, doi: 10.1002/jcp.22190, PMID: 20578247. In particular, the kidney is a very well vascularized tissue and kidney tumors, by mutation-driven VEGF-A overexpression, are well vascularized tumors, so Ren-PDOX models do not produce that much necrosis upon tumor growth as compared to other tumor types and implantation localizations. Experimentally, we have determined by sacrifice and tumor excision that our 12 mm (1000 mm³) orthotopic tumors have fully developed their vasculature to nurture the growing tissue and there is no necrosis in the tumor mass, as determined by H&E staining and Tunel staining of tissue sections (data not shown).

1st Sep 2020

Dear Dr. Casanovas,

Thank you for the submission of your revised manuscript to EMBO Molecular Medicine. We have now received the enclosed reports from the two referees who reviewed the new version of your manuscript. As you will see, they are now supportive of publication, and I am thus pleased to inform you that we will be able to accept your manuscript pending the following final minor amendments:

1) Please address referee #1's comments in writing in the manuscript.

2) Main manuscript text:

- Please answer/correct the changes suggested by our data editors in the main manuscript file (in track changes mode). This file will be sent to you in the next couple of days. Please use this file for any further modification.
- Please remove the red text.
- Please carefully check the text for spelling or grammatical mistakes (including the Material and methods section).
- We can accommodate a maximum of 5 keywords, please adjust accordingly.
- Please move the "Conflict of Interest" section further down, between the "Author contributions" and the "References".
- Please remove the "Statement of Significance".
- Material and Methods: As done in the checklist, please indicate the origin of cell lines and whether they were authenticated/tested for mycoplasma contamination. Please indicate the strain, origin, age, gender of mice, as well as the housing and husbandry conditions. Please include a statement confirming that informed consent was obtained from all subjects and that the experiments conformed to the principles set out in the WMA Declaration of Helsinki and the Department of Health and Human Services Belmont Report. Please indicate the antibody dilutions.
- Statistics: please indicate in the figures or in the legends the exact $n=$ and exact $p=$ values, not a range, along with the statistical test used. Some people found that to keep the figures clear, providing a supplemental table with all exact p -values was preferable. You are welcome to do this if you want to.
- The accession numbers and database should be listed in a formal "Data Availability" section (placed after Materials & Method). (see <https://www.embopress.org/page/journal/17574684/authorguide#dataavailability>). Please note that all links should resolve to a page where the data can be accessed before acceptance of the manuscript.

3) Figures:

A legend is missing for Fig. EV2C.

We would encourage you to include the source data for figure panels that show essential data. Numerical data should be provided as individual .xls or .csv files (including a tab describing the data). For blots or microscopy, uncropped images should be submitted (using a zip archive if multiple images need to be supplied for one panel).

4) Appendix:

Please add a Table of Content.

5) Checklist:

Section B/1: please detail "power studies were applied".

Section F: please provide accession numbers for the data generated in this study.

6) Thank you for providing a synopsis. I slightly modified your text to fit our style and format, please let me know if you agree with the following:

"Renal cell carcinoma (RCC) biopsy-derived orthotopic xenograft models (PDOX) reveal patient-specific induction of invasion and metastasis after antiangiogenics. Molecular characterization identifies ALDH1A3 as a pre-treatment discriminator of pro-malignant tumors that predicts response to therapy.

- Patient's original histomorphologic and molecular characteristics were maintained in an extensive series of 27 patient biopsy-derived orthotopic xenograft models (Ren-PDOX).
- Antiangiogenic treatment produced patient-specific responses of increased invasiveness and metastatic dissemination in approximately half of the models studied.
- By a novel technique of species-discriminative RNA-sequencing and subsequent filtering using patient data, the key molecular traits of pro-invasive type of tumors was unraveled.
- ALDH1A3 was clinically validated as a possible predictive factor of pro-aggressiveness in 15 antiangiogenic-treated RCC patients."

7) The paper explained: EMBO Molecular Medicine articles are accompanied by a summary of the articles to emphasize the major findings in the paper and their medical implications for the non-specialist reader. Please provide a draft summary of your article highlighting

8) As part of the EMBO Publications transparent editorial process initiative (see our Editorial at <http://embomolmed.embopress.org/content/2/9/329>), EMBO Molecular Medicine will publish online a Review Process File (RPF) to accompany accepted manuscripts.

In the event of acceptance, this file will be published in conjunction with your paper and will include the anonymous referee reports, your point-by-point response and all pertinent correspondence relating to the manuscript. Let us know whether you agree with the publication of the RPF and as here, if you want to remove or not any figures from it prior to publication.

I look forward to receiving your revised manuscript.

Yours sincerely,

Lise Roth

Lise Roth, PhD

Editor

EMBO Molecular Medicine

To submit your manuscript , please follow this link:

Link Not Available

The system will prompt you to fill in your funding and payment information. This will allow Wiley to send you a quote for the article processing charge (APC) in case of acceptance. This quote takes into account any reduction or fee waivers that you may be eligible for. Authors do not need to pay any fees before their manuscript is accepted and transferred to our publisher.

***** Reviewer's comments *****

Referee #1 (Remarks for Author):

Having gone over the revised manuscript as original reviewer #1, I am both gratified and satisfied that the authors have responded in a very meaningful way to address the various conceptual and technical concerns that I outlined previously. I won't comment on the revisions made regarding the technical points that I raised, but simply to say I think the changes have adequately addressed the issues raised. Regarding the conceptual issues, it was obviously good to see that the authors evaluated sunitinib, a first-line standard-of-care antiangiogenic tyrosine kinase inhibitor (TKI) drug for renal cell carcinoma (RCC), which they had not done in the prior version of the manuscript, where instead an antibody was used, that as a monotherapy, is not approved for the treatment of RCC patients. Second, the authors have added additional tumor models to their analysis and the results seem to be consistent with their prior findings. Furthermore, the issue about the apparent discrepancy between their preclinical outcome results, with the clinical outcomes observed in RCC patients treated with a drug such as sunitinib, or another TKI, seem reasonable - and address my concern. For example, they discuss how their results may help explain why such drugs often provide a benefit in progression free survival (PFS), but not overall survival (OS). This could be because of biologic pro-malignancy effects induced by the antiangiogenic drugs in some patients may contribute to this difference in PFS vs. OS outcomes.

I think the manuscript should now be accepted for publication. I think the only suggestion I would make for a minor revision (which does not have to be reviewed again) is to have a small section near the end which outlines some of the potential concerns or weaknesses of the overall study. This is now a standard procedure that is requested by the editors of a number of journals such as JNCI and Science Translational Medicine, and I like this policy. Thus, for example, I think one potential concern/weakness that remains is whether the increase in metastasis caused by antiangiogenic drug treatment seen in certain models reflects the fact that the treated mice simply survived for a longer period of time and as such, this may increase the probability of some mice eventually developing metastatic disease. This manuscript represents a tour de force analysis of a type that has rarely been duplicated by other investigators, and for that the authors should be commended.

Referee #2 (Comments on Novelty/Model System for Author):

This is a well presented and technically well done study.

Referee #2 (Remarks for Author):

the authors have provided an effective response to review. The revision is well done and addresses the concerns of the prior review. The inclusion of sunitinib as a therapeutic modality is useful. I enjoyed reading the revision and response to review.

The authors performed the requested editorial changes.

Corresponding Author Name: Oriol Casanovas

Manuscript Number: EMM-2019-11889